# A Survey on Data Selection for Language Models

**Alon Albalak,**\* *UC Santa Barbara, SynthLabs*

**Yanai Elazar,** *Allen Institute for AI, University of Washington*

**Sang Michael Xie,** *Stanford University*

**Shayne Longpre,** *Massachusetts Institute of Technology*

**Nathan Lambert,** *Allen Institute for AI*

**Xinyi Wang,** *UC Santa Barbara*

**Niklas Muennighoff,** *Contextual AI*

**Bairu Hou,** *UC Santa Barbara*

**Liangming Pan,** *UC Santa Barbara*

**Haewon Jeong,** *UC Santa Barbara*

**Colin Raffel,** *University of Toronto, Vector Institute*

**Shiyu Chang,** *UC Santa Barbara*

**Tatsunori Hashimoto,** *Stanford University*

**William Yang Wang,** *UC Santa Barbara*

**Reviewed on OpenReview:** *https://openreview.net/forum?id=XfHWcNTSHp*

## Abstract

A major factor in the recent success of large language models is the use of enormous and ever-growing text datasets for unsupervised pre-training. However, naively training a model on all available data may not be optimal (or feasible), as the quality of available text data can vary. Filtering out data can also decrease the carbon footprint and financial costs of training models by reducing the amount of training required.

Data selection methods aim to determine which candidate data points to include in the training dataset and how to appropriately sample from the selected data points. The promise of improved data selection methods has caused the volume of research in the area to rapidly expand. However, because deep learning is mostly driven by empirical evidence and experimentation on large-scale data is expensive, few organizations have the resources for extensive data selection research. Consequently, knowledge of effective data selection practices has become concentrated within a few organizations, many of which do not openly share their findings and methodologies.

To narrow this gap in knowledge, we present a comprehensive review of existing literature on data selection methods and related research areas, providing a taxonomy of existing approaches. By describing the current landscape of research, this work aims to accelerate progress in data selection by establishing an entry point for new and established researchers. Additionally, throughout this review we draw attention to noticeable holes in the literature and conclude the paper by proposing promising avenues for future research.

---

\*Work completed while at UC Santa Barbara. Correspondence to `alon_albalak@ucsb.edu`

**Table of Contents**

# 1 Introduction

Data selection is a long-standing challenge of machine learning where, given a collection of raw data, the goal is to design a dataset that is *optimal* under some objective function (John & Draper, 1975).

One often-used sense of optimality in data selection is with regard to a model's performance. In this work, we adopt the commonly held view that, at their core, *machine learning models are a method for modeling statistical patterns in data* and, from the probabilistic viewpoint, *the optimal dataset is that which most closely matches the distribution under which the model will be evaluated* (Murphy, 2012). While the probabilistic viewpoint presents a common view of how to select data that improves model performance, this is not the only goal of data selection methods. Data selection methods can reduce costs (by reducing dataset size (Ortiz Suárez et al., 2019; Schreiber et al., 2020; Brown et al., 2020a; Lee et al., 2022a; Sorscher et al., 2022)), ensure the integrity of evaluation metrics (by removing data that is suspected to be from the evaluation data (Rae et al., 2021; Marone & Van Durme, 2024; Oren et al., 2024)), and reducing undesirable behaviors (such as bias and toxicity (Dodge et al., 2021; Welbl et al., 2021; Luccioni & Viviano, 2021; Longpre et al., 2023c)).

Data selection has recently become especially important in the context of large language models (Zhao et al., 2023b; Minaee et al., 2024). Language models can undergo multiple stages of training (pretraining (Peters et al., 2018; Radford & Narasimhan, 2018; Devlin et al., 2019; Raffel et al., 2020; Touvron et al., 2023a), instruction-tuning (Mishra et al., 2021; Sanh et al., 2022; Longpre et al., 2023a; Muennighoff et al., 2024), alignment (Ziegler et al., 2019; Bai et al., 2022b; Ouyang et al., 2022; Rafailov et al., 2023), etc.), and data selection plays an important role in each stage. However, as the objectives of training differ across each stage, the goals of data selection also vary accordingly. For example, language models are commonly pretrained on large corpora of text from a variety of sources. Among these sources is Common Crawl, a collection of around 250 billion webpages scraped from the internet. These webpages amount to around 11 petabytes of data, collected from internet scraping efforts since 2008, with an additional 3-5 billion new web pages being crawled monthly.[1] Due to the massive size of pretraining corpora, a common goal of data selection during pretraining is to remove significant quantities of data through a series of filters (Conneau & Lample, 2019; Raffel et al., 2020; Wenzek et al., 2020; Gao et al., 2020; Rae et al., 2021; Lee et al., 2022a) that aim to only retain data that is deemed "high-quality". In contrast with pretraining, one form of data selection for fine-tuning a model on a target task is to select additional auxiliary samples that will be most beneficial as additional learning signals for the target task (Albalak et al., 2023b; Ivison et al., 2023a).

In this work, we unify the wide array of data selection methods under a conceptual framework that allows us to compare and contrast the variety of methods under our probabilistic viewpoint (§2.2) with a focus on model pretraining. Through this survey, we demonstrate how all data selection methods define a *utility function* that determines the utility of data, as well as a *selection mechanism* that determines how to use a data point based on its utility. Our conceptual framework enables us to classify the wide variety of data selection methods and create a taxonomy of the existing literature. Overall, this survey aims to achieve two goals: 1) we provide a collected resource of data selection methods that describes the current best practices and considerations when selecting data for training a language model, and 2) we provide a unifying view of data selection methods that allows us to define and describe potentially fruitful future directions of research. While this survey aims to be comprehensive, it would be prohibitively long to include the exact details of every method, so we select a representative sample to discuss in depth and provide citations to the many methods that we cannot cover in depth.

We organize the survey as follows. First, we present a taxonomy and unified conceptual framework for data selection (§2). Next, we present the main focus of this work: surveying methods of data selection for language model pretraining (§3). Then, we follow up with data selection methods for other language model training regimes including multitask training and instruction-tuning (§4), alignment (§5), in-context learning (§6), and task-specific fine-tuning (§7). We then extend the survey to notable methods of data selection used in domains other than language (§8), as well as a brief discussion of related topics (§9). Then, we discuss some of the implications surrounding data selection (§10). Although the main focus of the survey is on language model pretraining, we also cover methods from other training regimes and domains so that we can point out promising directions of research in the final section (§11).

---

[1]According to https://commoncrawl.org/ accessed on 11/02/23 (MM/DD/YY).

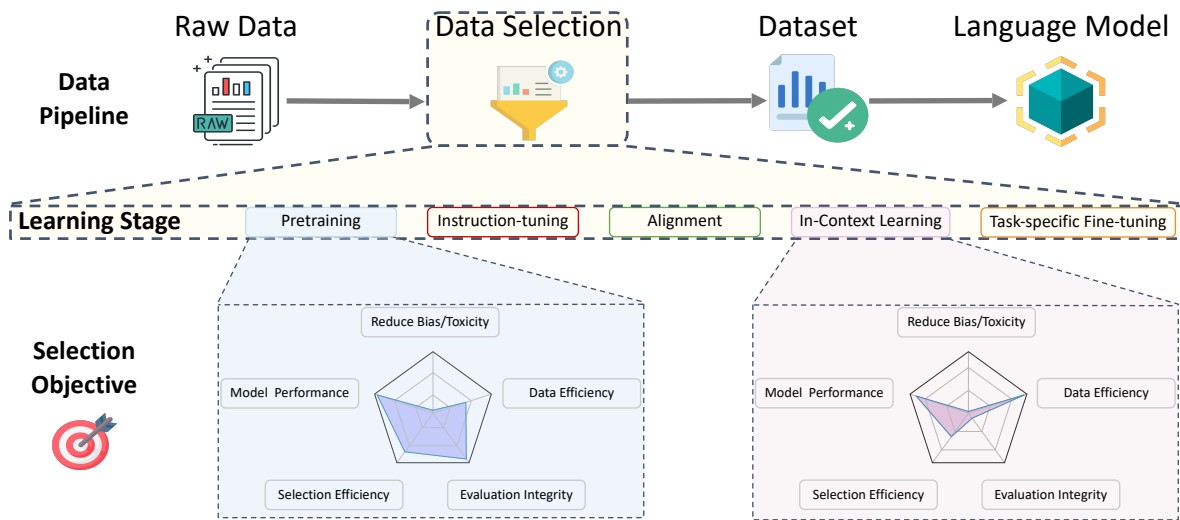

Figure 1: **An overview of the data pipeline for language models.** The process starts with raw data that is cleaned, filtered, and mixed to create a final dataset by the data selection process, then used to train (or evaluate) a model. The details and objectives of data selection methods vary depending on the learning stage, and we identify five common *objectives*: improving model performance, improving data efficiency, selecting data efficiently, ensuring evaluation integrity, and reducing model bias and toxicity. For example, when selecting data for pretraining (§3) we may care about data and selection efficiency more than about reducing toxicity in the data. In addition to pretraining, data selection can be used for instruction-tuning (§4), alignment (§5), in-context learning (§6), and task-specific fine-tuning (§7), where each stage has differing priorities.

## 2 A Taxonomy for Data Selection

In this section we define our taxonomy of data selection methods. We start by defining concepts for data points and datasets (§2.1), then we describe the components of a data selection method in our unified framework (§2.2), followed by common dimensions of variance in the taxonomy (§2.3).

### 2.1 Background and Motivation of Data Selection

Prior to discussing data selection methods, we first define and describe the "units" of a dataset, from smallest to largest. First, the smallest unit of data for language models is the token, which can be composed of bytes, characters, subwords, or larger components, depending on the tokenizer.[2]

**Definition 2.1** (*Data point*)**.** A data point, $x^{(i)}$, is an ordered collection of tokens that constitutes a single sample of data used to train or evaluate a language model.

For example, in language modeling, $x^{(i)}$ can be a sequence of tokens from an internet document, a book, or a scientific article. Language models often have a limited input sequence length when training, so long documents are often split into multiple data points. In practice, this means that $x^{(i)}$ may constitute a portion of a document (rather than a complete one).

**Definition 2.2** (*Data point characteristics*)**.** The characteristics of a data point refer to a number of measures used to describe a data point, $x^{(i)}$, and determines where $x^{(i)}$ is located within the space of all possible data points.

Common measures used to describe data points can be either simple statistics (*e.g.*, number of characters in $x^{(i)}$, or the percent of alphabetic characters in $x^{(i)}$) or a distributed representation (*e.g.*, the embedded

---

[2]How to effectively encode texts is still an active research area (Rust et al., 2022), but to date splitting text into tokens is the leading approach.

representation using BERT (Devlin et al., 2019)). The data point characteristics are a crucial component of data selection as they are frequently used to determine whether a data point should be cleaned or possibly removed entirely.

**Definition 2.3** (*Dataset*). A dataset, $\mathcal{D}$, is the collection of data points $\{x^{(1)}, \ldots, x^{(N)}\}$ (where $N = |\mathcal{D}|$ is the number of examples in $\mathcal{D}$) that will be used to either train or evaluate a model.

**Definition 2.4** (*Dataset distribution*). The distribution of a dataset, $\mathcal{D}$, refers to the distribution of data points within the space of all possible data points. The distribution of $\mathcal{D}$ has a significant impact on the final capabilities and properties of a model that is trained on it.

The dataset distribution is also a crucial component of data selection as models can struggle with out-of-distribution generalization and the dataset distribution specifies what data is in-distribution vs. out-of-distribution (Shi et al., 2023). Specifically, a dense region of the data distribution generally suggests that a model will perform well on unseen data from that region (*e.g.*, similar distribution of tokens). With this in mind, increasing the density of data around desirable regions while reducing density around undesirable regions generally leads to a model that performs well in the desired settings.

## 2.2 A Unified Conceptual Framework for Data Selection

**High-level goal of data selection.** Data selection is the process of creating a dataset from a collection of candidate data points, which will be used to train or evaluate a machine learning model. Prior to data selection, generally speaking, data is collected, (optionally) annotated, and stored. Then, once the target use case for a model is determined, *the goal of data selection is to filter and select the data points that maximize a desired objective.*

**Formal definition of data selection.**

**Definition 2.5** (*Data selection*). A data selection function, $\phi$, takes as input a dataset $\mathcal{D}_{\text{raw}}$ and objective function $f_{\text{obj}}$ and filters, cleans, and selects data points from $\mathcal{D}_{\text{raw}}$ to create a final dataset $\mathcal{D} = \phi(\mathcal{D}_{\text{raw}})$ such that, for a model $\mathcal{M}$ trained on dataset $\mathcal{D}$, $\phi$ aims to maximize/minimize the objective function $f_{\text{obj}}(\mathcal{M})$.

In practice, multiple selection methods are often composed to improve the coverage and specificity of the final dataset. In this case, we denote the component functions as $\phi_j$ and the composition as $\phi(\mathcal{D}) = \phi_1 \circ \cdots \circ \phi_n(\mathcal{D})$. For example, $\phi = \phi_1 \circ \phi_2 \circ \phi_3$ can include a simple filtering component, $\phi_3$, that first removes undesirable data points, followed by a cleaning component, $\phi_2$, that removes undesirable content from within individual data points, followed by a mixing component, $\phi_1$ that determines the number of times the remaining data points should be used in the final dataset. Through these mechanisms, data selection functions can adjust both the individual *data point characteristics* and the *dataset distribution* towards more desirable regions of the data space in the aim of improving desirable properties of the model for the target purpose.

**Components of a data selection method.** We identify the common components of selection functions $\phi_j$: *the utility function* and *selection mechanism.*

The **utility function** $\upsilon(x^{(i)}) : \mathcal{D} \to \mathbb{R}$ defines a mapping from a data point to a real number representing the calculated utility. For example, a utility function may be a binary indicator defined as: 1 if the total number of tokens in $x^{(i)}$ is greater than 10, or 0 otherwise. Another utility function might assign $x^{(i)}$ a utility equal to the likelihood of $x^{(i)}$ being a Wikipedia article.

The **selection mechanism** uses the output from the utility function to determine whether a data point will be included in the resulting subset (and in some cases, how many times the data point should be repeated). It can be a simple indicator (*e.g.*, only include $x^{(i)}$ in $\mathcal{D}$ if the number of characters is greater than 10), or probability functions (*e.g.*, if $\upsilon$ is the likelihood of $x^{(i)}$ being a Wikipedia article, include $x^{(i)}$ in $\mathcal{D}$ according to the probability defined by $\upsilon(x^{(i)})$). Additionally, a *selection sensitivity* is needed for selection mechanisms that require a threshold (*e.g.*, only include $x^{(i)}$ if $\upsilon(x^{(i)}) > 0.9$).

Data selection methods are used to solve a wide range of objectives, and by adjusting the utility mechanism, selection mechanism, and filter sensitivity can achieve the desired outcomes. We save discussion of the specific instantiations of each component to the respective sections where they are introduced.

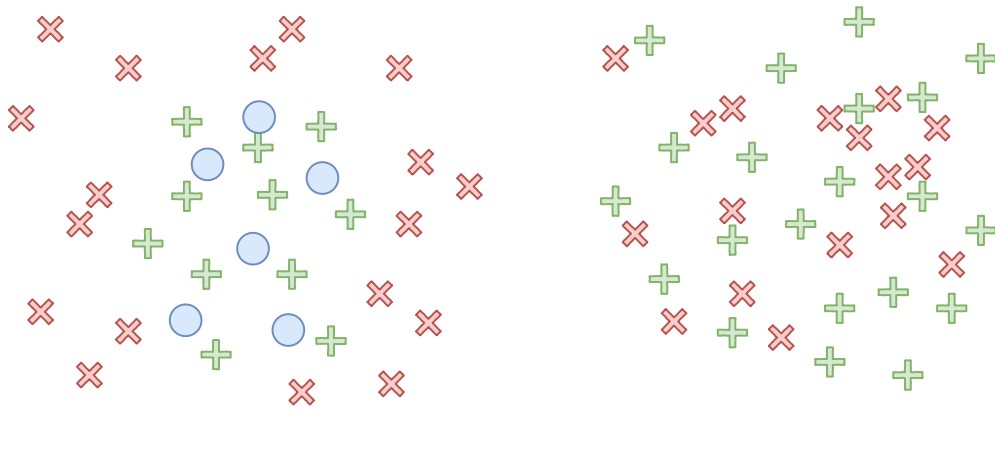

Figure 2: **A conceptual demonstration of two common goals for data selection methods: Distribution Matching and Distribution Diversification.** On the left we see distribution matching where the goal is to select data points (green crosses) that are similar to data points sampled from a target data distribution (blue circles) and reject data that is too far out of distribution (red exes). On the right we see dataset diversification which aims to select/reject data points (green crosses/red exes) in such a way that maintains coverage over the full distribution while reducing the total number of data points.

## 2.3  Dimensions of Variance in the Data Selection Taxonomy

Data selection methods can be utilized for various goals, where the goal of the method will, in part, determine the exact utility function and selection mechanisms used. To help form a taxonomy and a better understanding of the relationship between methods, we define some specific dimensions of commonality and variance across methods (in no particular order).

### 2.3.1  Distribution Matching vs. Diversification

The main goal of **distribution matching** (Figure 2, left) methods is to select data with properties similar to the desired target distribution, upon which a model will be evaluated or deployed. For instance, the desired distribution could be defined as data of known high quality, a specific language, or a target domain (*e.g.*, finance, medicine, or law). The exact specifications of the desired target distribution can vary from being well-defined (*e.g.*, as in detecting a language) to being quite ambiguous (*e.g.*, "high-quality" data). Some distribution matching methods will try to match data representation distributions where the similarity to the target distribution will usually be the utility (*e.g.*, similarity to Wikipedia data). Other distribution matching methods use the statistics of data sampled from the target dataset as the utility (*e.g.*, total number of characters per example).

**Distribution diversification** methods (Figure 2, right) aim to prioritize heterogeneity in a sample, removing redundancies. They operate within a representation space that allows for similarity to be measured across data points. The utility of a data point is defined by its relation (similarity) to the other data points in the representation space. Distribution diversification methods often remove data points that are similar in some representation space (*e.g.*, characters or vectors). By removing data points that are very similar, diversification methods can remove redundancies in the data, leading to improved training efficiency by reducing the dataset size. Additionally, because distribution diversification methods force the dataset distribution assign probability to a broader range of datapoints, they can lead to decreased memorization, decreased bias, and improved robustness.

### 2.3.2  Altering the Dataset vs. Data Point

Methods that **alter the dataset** aim to increase or decrease the frequency of individual data points within the dataset in order to increase (decrease) the resultant distribution density around desirable (undesirable) regions. These methods take as input a dataset and assign each individual data point a non-negative integer value representing the number of times that data point should be included in the resultant dataset. Formally, a dataset distribution altering function $\phi_j : \mathcal{D}^N \to \mathbb{N}_0^N$ maps each data point $x^{(i)} \in \mathcal{D}$ to a non-negative integer based on the utility function, $v(x^{(i)})$. Each datapoint is then filtered ($\phi_j(x^{(i)}) = 0$), maintained ($\phi_j(x^{(i)}) = 1$), or oversampled ($\phi_j(x^{(i)}) > 0$) according to the utility function.

Methods that **alter the data point** aim to adjust the content within a data point to better match a desirable distribution of tokens so that the individual data points within a dataset more closely resemble the desired characteristics. In practice, this can take the shape of removing individual lines, or chunks of text, from a document. For example, when training a model for natural language, it can be beneficial to remove HTML tags as they are likely to be out of distribution if the desired target domain is natural language.

### 2.3.3  Output Space: Binary vs. Natural Number Selection

Methods that alter the dataset distribution lead to another dimension of variance: whether the selection function $\phi$ assigns binary values (*i.e.*, include or remove) or whether larger integer values are permitted. Methods that only **assign binary values** are generally referred to as filtering, and the goal is usually to adjust the dataset distribution by removing undesirable data points from the dataset. On the other hand, methods that can **assign any natural number** are commonly referred to as data mixing, and their goal is often to adjust the dataset distribution by prioritizing certain subsets of the dataset that have high utility values, while decreasing the distribution density around subsets with lower utility. It is very common for data mixing methods to define a utility function that assigns the same utility to entire subsets of data (*e.g.*, all web text gets the same utility, and all books get a different utility). Filtering and mixing can be used as part of a pipeline, where data is first filtered and then mixed.

### 2.3.4  Training Stage

Language model training often requires multiple stages where each stage serves a different purpose. In this work we identify and discuss five distinct language model training stages during which data selection can be used: **pretraining, instruction-tuning, alignment, in-context learning,** and **task-specific fine-tuning**.

Each training stage has different goals, and so the data selection methods for each stage will use different mechanisms to achieve those goals. For example, the target distribution is fairly well-defined when training a language model for a specific domain or task, but much less defined when pretraining a general purpose language model, thus data selection methods can utilize this information to target better data. Another consideration is the number of candidate data points at each training stage. For example, pretraining likely has a significantly larger number of candidate data points than instruction-tuning, motivating the need for efficient data selection methods in pretraining, while instruction-tuning can afford more expensive methods.

Data selection has been studied, to some extent, in all stages of training. However, information on data selection for pretraining is most limited, and therefore we dedicate the main portion of this work to comparing, contrasting, and better understanding the methods of data selection in pretraining, with more concise sections discussing data selection for other training stages.

*Pretraining Data Selection Methods*

| Selection Method | Distribution Matching vs. Diversification | Output Space | Adjust Dataset vs. Data Point |
|---|---|---|---|
| Language Filtering (§3.1) | M | $\{0,1\}$ | D |
| Heuristic Approaches (§3.2) | M | $\mathbb{N}_0$ | D + P |
| Data Quality (§3.3) | M | $\{0,1\}$ | D |
| Domain-specific (§3.4) | M | $\{0,1\}$ | D |
| Deduplication (§3.5) | D | $\{0,1\}$ | D + P |
| Toxic and Explicit Content (§3.6) | M | $\{0,1\}$ | D + P |
| Multilingual Filtering (§3.7) | M + D | $\mathbb{N}_0$ | D + P |
| Data Mixing (§3.8) | M + D | $\mathbb{N}_0$ | D |

Table 1: Pretraining data selection methods can be described along three axes of variation: whether they aim to do distribution matching or diversification, what the output space is, and whether they make adjustments to the dataset distribution or data point characteristics.

## 3 Data Selection for Pretraining

The goal of pretraining is usually to train a "general-purpose" model, which requires training on massive quantities of text (often trillions of tokens for recent large language models). Selecting the best data from such large quantities can be very expensive, so a common first step in the process is to remove data with various filters, and it is common to apply multiple filters that are pipelined together to achieve the desired dataset. The order in which we present pretraining data selection methods is, roughly, based on the order that they are used within real data selection pipelines. Of course, not all pipelines require every method presented here, and depending on the case the exact ordering may differ slightly. We provide a high

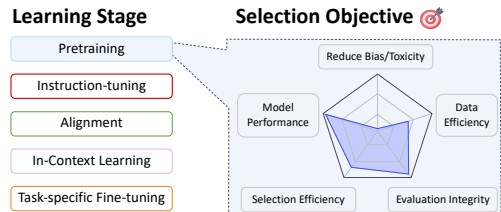

Figure 3: **Data selection: pretraining**. The first training stage of modern language models. Typically, the most important selection objectives are *model performance*, *evaluation integrity*, and *selection efficiency*.

level overview of the filtering pipline in Figure 4, and references to relevant subsections as well as axes of variation in Table 1. The pretaining data selection step illustration is depicted in Figure 3.

### 3.1 Language Filtering

When curating data for language model pretraining, a crucial first step is to consider the languages the model will operate in and to filter out data that doesn't belong to those languages. This applies not only to natural languages, but coding languages as well, however, the methods to determine each differ in practice. For multilingual language models, in addition to filtering out undesired languages, it is also important to track metrics on the quantity of data coming from each language. Similarly, for models with code capabilities, it is important to track the quantity of data from each coding language.

**Common utility functions.** When filtering for language, it is crucial that the utility functions be fast to compute as the utility calculation is often performed on all available raw text data (potentially a huge quantity). Thus, many methods that aim to filter for specific natural languages utilize fast-to-compute utility functions from classifiers based on character n-grams (Conneau & Lample, 2019; Wenzek et al., 2020; Raffel et al., 2020; Xue et al., 2021; Laurençon et al., 2022) including LANGDETECT[3], CLD3[4], and FASTTEXT (Joulin et al., 2016; Grave et al., 2018). Some methods aim to remove all non-English data, while others may include over 100 languages and require different utility function strategies.

---

[3]https://pypi.org/project/langdetect/
[4]https://github.com/google/cld3

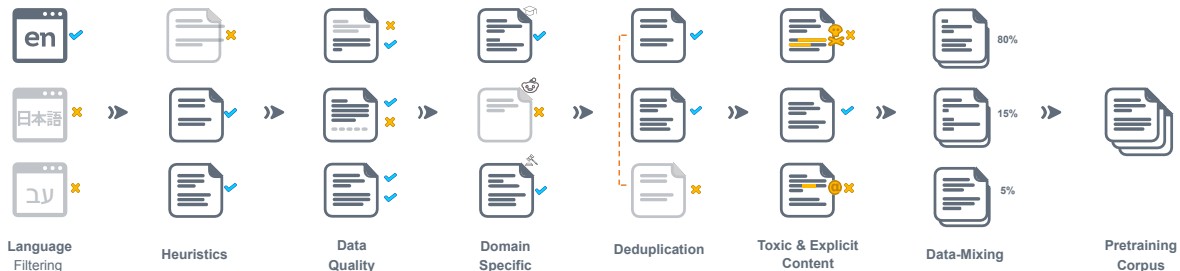

Figure 4: **An overview of the data filtering pipeline for pretraining**. Each filtering component is described in Section 3, and depicts common filters used for preprocessing text data. Note that different works employ different filters, at different stages, and do not necessarily adhere to the order conveyed here. This figure was adopted and modified from Soldaini et al. (2024).

**Filtering for English-only datasets.** When developing the English-only `C4`, Raffel et al. (2020) use a naive Bayes classifier with character n-grams (LANGDETECT) to filter out any pages not classified as English with a probability of 0.99. They find the use of a very high threshold sensitivity to be appropriate as the filtered dataset still contains a significant quantity of data (~750GB of data). More recently, when developing `Dolma`, Soldaini et al. (2024) utilize FASTTEXT's (Grave et al., 2018) language identification model and filter out any documents classified as English with a probability less than 0.5, finding that this removed 61.7% of web pages. However, when filtering books (from Project Gutenberg[5]) they first split each book by paragraph and compute the language score for each paragraph, only removing books that have an average probability of English (assigned by FASTTEXT) of below 0.5. Penedo et al. (2023) also use FASTTEXT when filtering `RefinedWeb`, but differently to `Dolma`, they use a higher threshold (0.65) and therefore filter out higher quantities of web pages.

The lower thresholds used by more recent methods are likely due to a combination of factors. First, ever-growing model sizes have necessitated larger pretraining datasets, and using a lower threshold will allow more data to pass through the filter. Additionally, it is difficult to measure the differences in accuracy across different language identification models (*e.g.*, LANGDETECT vs. CLD3. vs. FASTTEXT) due to the differences in supported languages and strengths on text of varying lengths, but some experiments show that FASTTEXT has improved accuracy over other language identification models,[6] as well as reduced latency. In general, the threshold used for keeping or filtering out a data point can be in part determined by the desired dataset size, where a lower threshold will retain higher quantities of data, but possibly containing non-English text.

**Filtering for multilingual datasets.** Filtering methods for multilingual corpora often rely on the FASTTEXT (Joulin et al., 2016) language classifier from CCNET (Wenzek et al., 2020) which was trained to classify 176 languages using wikipedia data (Grave et al., 2018; Conneau et al., 2020; Laurençon et al., 2022). The FASTTEXT classifier is desirable as it can process 1,000 documents per second on a single CPU core (Wenzek et al., 2020). One example of this method comes from Laurençon et al. (2022) who use a FASTTEXT model to obtain a prediction for the document language along with a prediction confidence score when creating the `ROOTS` corpus. If the confidence score is below a threshold, the document is removed, where the confidence score is determined by native speakers on a language-by-language basis, but are not disclosed in their paper. The need to determine a threshold for each language individually adds significant amount of additional effort to develop language filters for multilingual corpora. Additionally, these filters can fail on documents in which the language changes several times as the classifier cannot confidently predict a single language. Furthermore, while this classifier was trained on 176 languages, there are 1000s of languages that cannot be covered by this method (van Esch et al., 2022). This has prompted the creation of even broader language detectors covering thousands of languages, however, they still suffer from poor precision for extremely low-resource languages (Caswell et al., 2020; Kudugunta et al., 2023).

---

[5]https://www.gutenberg.org/
[6]https://modelpredict.com/language-identification-survey

Another common approach for multilingual filtering is by country domains or selecting URLs that are known to contain data in certain languages (Luukkonen et al., 2023). Especially for very low-resource languages, such as Uyghur, this can be a more reliable method than language classifiers (Zhang et al., 2023a).

For further discussion on data selection for multilingual models, see Section 3.7.

**Filtering for code languages.** The utility functions that have been used to detect code languages are relatively simple compared to those used for natural languages. For example, Chowdhery et al. (2023) filter for data from 24 programming languages simply by searching for documents that match a set of approved filename extension (*e.g.*, ".py" for python). This strategy works well when filtering over data from known code data that contains file names, such as snapshots of Github repositories (Chen et al., 2021; Li et al., 2022a). However, no filtering methods that we know of find code within natural language documents, which is entirely possible within some domains such as in Stack Exchange[7] and may contribute meaningful quantities of interleaved natural language and code.

**Challenges for language filtering.** There are trade-offs occurring in the filter sensitivity for language detection, where a lower threshold allows greater quantities of data to pass through the filter at the risk of including lower quality data. On the one hand, using a strict filter for English data may reduce the quantities of non-English data, but on the other hand, using a lower threshold can mitigate some of the inherent biases of the language detector against dialects spoken by minority groups (*e.g.*, African-American Vernacular English (Blodgett et al., 2016)). Determining the exact threshold to use for language classifiers is highly dependent on the use case. While FASTTEXT is the current standard for language detection due to its combination of speed and accuracy, it is not perfect. More accurate systems built in on recurrent- (Toftrup et al., 2021) and transformer-based[8] architectures have been developed, but run with higher computational costs.

---

**Language Filtering Summary & Ideas**

- Language filtering is an important first step of data selection where documents are selected to include only the desired languages.
- Classifier-based methods are the norm when creating both english-only and multilingual datasets.
- For lower-resourced languages, URL-based methods are also useful for classifying the language used in multilingual datasets and can sometimes can be more reliable than classifiers.
- Filtering for code languages is very simple with most works simply using the file extension.
- 💡 One interesting direction for future research is developing methods that find code data within documents of natural language. To the best of our knowledge, no such methods currently exist, but a code-language identifier could be used in domains with interleaved code and natural language to find meaningful quantities of new code data.

---

[7]https://archive.org/details/stackexchange
[8]https://huggingface.co/papluca/xlm-roberta-base-language-detection

| Heuristic Category | Common Utility Functions | Example Selection Mechanisms |
|---|---|---|
| **Item Count** | # of characters in a {word/line/paragraph/document} 
 # of {words/lines/sentences} in a document | Remove documents with fewer than 5 words (Raffel et al., 2020) |
| **Repetition Count** | # of times a {character/n-gram/word/ sentence/paragraph} is repeated | Remove lines that repeat the same word more than 4 times consecutively (Laurençon et al., 2022) |
| **Existence** | Whether a {word/n-gram} is in the document 
 Whether a terminal punctuation is at the end of a line | Remove lines starting with "sign-in" (Penedo et al., 2023) |
| **Ratio** | % of alphabetic characters in a document 
 % of numerals/uppercase characters in a {line/document} | Remove documents with a symbol-to-word ratio greater than 0.1 (for "#" and "...") (Rae et al., 2021) |
| **Statistics** | The mean length (and standard deviation) of all lines in a document | Remove code files that have mean line length greater than 100 characters (Chen et al., 2021) |

Table 2: **Commonly used heuristic utility functions and demonstrative selection mechanisms.**

## 3.2 Heuristic Approaches

When pretraining a language model, raw datasets are generally composed of massive quantities of text from a wide variety of sources that are filtered through a set of simple heuristics. Major models are often trained on web scrapes such as CommonCrawl and GitHub, though transparency into their precise compositions are on the decline (Bommasani et al., 2023). It is widely known however that raw text data from the internet can contain significant quantities of boiler-plate text, error messages, and offensive text (Raffel et al., 2020; Touvron et al., 2023a; Elazar et al., 2024). For example, Gao et al. (2020) find that, when creating the Pile, the most common 13-grams were character repetitions such as a string of dashes ("– –") with 11 million instances. Removing such undesirable text is very important, but must be done efficiently due to the size of the corpora involved. In this case, a common approach to filtering data involves simple and efficient-to-compute heuristics.

The goal of heuristic approaches is to constrain the training distribution along some dimension (*e.g.*, sentence length, repetitiveness), with the assumption that the evaluation distribution will exhibit similar characteristics. The number of heuristics that have been used in past works is extensive, but generally fall into one of the following categories of heuristics: **item count, repetition count, existence, ratio,** or **statistics**. In this section we discuss a representative sample of them, provide Table 2 as an overview of heuristic methods, and include a comprehensive listing in Appendix A.

**Common utility functions.** For heuristic-based text filtering, the utility functions should be very fast to compute and generally rely on one or more qualities of the raw text included in each data point of the dataset. However, the exact text qualities that lead to best performance depend on the desired use case for the model, and are yet unknown. Thus, a wide variety of distinct utility functions have been proposed, each making assumptions about the expected testing distribution, and filtering out data that does not fit those assumptions. For example, Raffel et al. (2020) discard any web page with fewer than five sentences, Rae et al. (2021) discard documents with fewer than 50 or greater than 100,000 words, and Xue et al. (2021) discard documents with fewer than three lines of text with 200 or more characters. Another common heuristic is to remove documents with specific black-listed words or phrases. Raffel et al. (2020) remove documents with any word on the "List of Dirty, Naughty, Obscene, or Otherwise Bad Words".[9] However, Penedo et al. (2023) suggest that the NSFW word block lists as used by Raffel et al. (2020) can often lead to false positives, resulting in the over-filtering of legal and medical content.

---

[9]https://github.com/LDNOOBW/List-of-Dirty-Naughty-Obscene-and-Otherwise-Bad-Words

Another heuristic often associated with the quality of a document is repetitiveness. Rae et al. (2021) find that excessive repetition within a document is linked to uninformative content and suggest that filtering out entire documents that have high proportions of repeated lines, paragraphs, and n-grams leads to improved performance. For example, if a repeated line accounts for greater than 30% of a document they remove the entire document. See Table A1 in Rae et al. (2021) for their list of 13 different repetitions and threshold sensitivities.

**Domain-specific heuristic approaches.**   Another dimension to consider when designing a filter is the domain of interest. For example, when collecting code data for the ROOTS dataset, Laurençon et al. (2022) filter source files between 100 and 200,000 characters, with between 15-65% alphabetic characters (a-zA-Z), and line lengths between 20-1000 characters. If these filters were applied on natural language data, significant quantities of desirable data would be removed (*e.g.*, many books have greater than 65% alphabetic characters). Additionally, Laurençon et al. (2022) filter out files that have line lengths that are highly uniform (line length standard deviation 3 characters or less). While filtering out data with uniformly lengthed lines may improve data quality for code, it would likely filter out significant quantities of poems and other rhythmic text due to the similarity in line length, and would likely remove data that has been collected from pdf files, which could be desirable when training a model for a non-code domain.

**Adjusting data points with heuristics.**   While the previously described approaches remove entire data points from the training dataset, removing content from within a data point can be a useful method. Rather than removing an entire data point, heuristic methods that remove individual lines within a larger document aim to improve the coverage of the training data distribution, while limiting the density around undesirable qualities. For example, when working with content from web crawled data, removing HTML can be an important part of the filtering process that helps to reshape the data distribution towards natural language (assuming that is the desired outcome). Laurençon et al. (2022) and Aghajanyan et al. (2022) create lists of HTML tags where all content within sub-trees of the tags will be removed. Additionally, Raffel et al. (2020) remove lines that do not end in a terminal punctuation mark and lines with fewer than four words, and Penedo et al. (2023) remove lines that are mainly composed of uppercase or numerical characters. Furthermore, a common component of line-by-line filtering is the removal of the entire data point if enough of the individual lines within it are filtered out. For example, Penedo et al. (2023) remove the full document if the individually removed lines account for greater than 5% of the total document length.

**Deterministic vs. stochastic selection mechanisms.**   Many heuristic utility functions can be used as part of a deterministic or stochastic selection mechanism. In a deterministic selection mechanism, any data points which are above/below the utility threshold will be removed. The vast majority of existing heuristic filters use a deterministic selection mechanism. However, this may unnecessarily remove data that resembles the evaluation distribution (Brown et al., 2020a). In particular, the target dataset distribution may be long-tailed in the characteristic of interest, and deterministically removing all data above/below a threshold can harm the modelling of that characteristic.

Heuristic filters commonly based on item counts, repetitions, ratios, and statistics can all utilize stochastic selection mechanisms. For example, Rae et al. (2021) remove documents with a symbol-to-word ratio greater than 0.1, but a stochastic selection mechanism would allow for *some* data with low utility to not be filtered out. While a stochastic selection mechanism may not always be appropriate, there are certainly some heuristics where it may benefit the dataset distribution, but the exact settings where a stochastic selection mechanism is an open research question. Data quality filters (§3.3) often use stochastic selection mechanisms, and applying stochasticity in heuristic filters is one area of data selection that has not been explored.

**Drawbacks of heuristic selection approaches.**   One disadvantage to heuristic selection approaches is that they do not judge the quality or content of a document, rather, they rely entirely on surface level statistics and counts, lacking the "finesse" of other data selection methods. This can lead to desirable data being removed, but this is often accepted as a trade-off as long as the quantity of data after filtering is still within the desirable range.

Another disadvantage of heuristic filters is the challenge of validating their effectiveness. After a heuristic has been developed and used on a raw dataset, to determine whether the heuristic is an improvement requires either: 1) manually validating that the filtered documents are undesirable and the kept documents are desirable, or 2) training and evaluating a model. Both of these methods are time consuming and expensive, which make experimentation on heuristic filtering approaches very slow.

**Considerations when designing a heuristic filter.** When designing a set of heuristic filters, it is very important to understand the data source. By first looking at the data and validating the performance of a heuristic filter on a small sample of data, the designer of such a filter can better understand what type of data the method will remove, and what types of data will remain. That way, by composing multiple heuristics in succession, the resultant dataset will reflect the desired data qualities. There are a wide variety of existing tools to better understand the data within a dataset (data auditing), as well as for data selection. See §10.4 for some useful tools.

The current known best practices for heuristics have not changed significantly in the past few years. Soldaini et al. (2024) recently use a combination of the filters from `MassiveText` (Rae et al., 2021) and `C4` (Raffel et al., 2020), while Penedo et al. (2023) also follow the rules defined for `MassiveText` (Rae et al., 2021). However, it is unclear whether the lack of progress in common heuristics is due to a lack of exploration in the space (due to limited resources and studies), or if the known heuristics truly are optimal. To the best of our knowledge, there have not been studies specifically comparing the many possible heuristics in a systematic study.

---

**Heuristic Approaches Summary & Ideas**

- Heuristics are a very efficient method for removing large quantities of data, but lack finesse.
- Most common heuristic utility functions fall into 5 categories: item count, repetition count, existence, ratio, and statistics.
- Heuristics should be designed specifically for the domain of interest (*e.g.*, heuristics for code may not work for natural language).
- Heuristics can also be used to improve data point characteristics (*e.g.*, removing HTML from a web document, or specific lines within a document).
- 💡 Because heuristics lack finesse, it may be worth exploring the use of stochastic selection mechanisms to allow higher quantities of data through the filter (with the hope that a filter farther down the pipeline will remove data that is actually undesirable).
- 💡 Validating the quality of a heuristic filter is a very slow process, requiring either manual inspection, or training and evaluating a model. Developing methods that can directly measure the data itself could significantly improve iteration time.

### 3.3 Data Quality

Training on the highest quality data can lead to stronger performance (Du et al., 2022). However, "high-quality" is not a well-defined term with respect to language model pretraining data. Additionally, there is no one-size-fits-all characterization as, depending on the use case, "high-quality" data can vary drastically (Longpre et al., 2023c). Additionally, implementations of quality filters (e.g. for detecting toxicity) can further marginalize communities whose text is not commonly considered "high quality" (Xu et al., 2021; Longpre et al., 2023c). In this work, we narrow the use of the phrase "high-quality" to a common use of the phrase referring to data that is known to be written by humans and has likely gone through an editing process (Grave et al., 2018; Gao et al., 2020; Brown et al., 2020a; Chowdhery et al., 2023). Some data domains that fall under the "high-quality" category are Wikipedia, books, patents, and peer-reviewed journal articles.

**Common utility functions.** Similarly to heuristic methods, the utility functions used for data quality filtering should be fast and cheap to compute, however, in contrast with heuristic filtering approaches, quality filtering requires fuzzy matching which generally has higher computational requirements. Thus, methods for data quality often use relatively cheap distributional representation methods, such as n-grams, to allow for more ambiguous filtering criteria.

One commonly used method when filtering for quality is **classifier-based quality filtering**, where the goal is to identify data points that are likely from the same (or similar) distribution as a known "high-quality" corpus of data points (reference corpus) (Longpre et al., 2023c). For example, Brown et al. (2020a) filters a version of Common Crawl[10] based on similarity to "high-quality" reference corpora (including `WebText` (Radford et al., 2019a), `Books1`, `Books2`, and English wikipedia). They train a classifier using the "high-quality" reference corpora as the positive class and unfiltered Common Crawl documents as the negative class. This classifier then defines the utility metric for selecting data that follows the desired distribution. The classifier used to assign quality scores to each document should be fairly lightweight due to the scale of pretraining datasets. For example, Du et al. (2022) use a feature hash based linear classifier, while Brown et al. (2020a) and Gao et al. (2020) use a FASTTEXT model (Joulin et al., 2016) with an n-gram size of 2. Similarly, Xie et al. (2023b) computes the utility as an importance weight between two hashed n-gram generative models. While modern language models may be able to better model the reference corpus distribution, they would also significantly increase the amount of compute required over hashing and n-gram based methods.

A competing method of computing utility is **perplexity-based quality filtering**, where the goal is to train a language model on the reference corpora and evaluate on the data to be filtered, assigning a utility score to each candidate document. For example, Wenzek et al. (2020) train a 5-gram KNESER-NEY (Heafield, 2011) model on Wikipedia, and compute the perplexity of each paragraph in their Common Crawl dump. Paragraphs with low perplexity are assumed to be close to the reference domain while data with paragraphs with higher perplexity are assumed to be of low quality.

Recently, Wettig et al. (2024) have explored a very different direction, and create a utility function from the outputs of a language model system (GPT-4) by asking it to rate multiple documents along various dimensions of perceived quality. While their work shows that the model-based ratings can be a good signal for quality, it can be costly to make enough API calls to such systems. Additionally, biases from the system may be carried over into the quality signal.

**Selection mechanisms.** One potential negative impact of classifier-based quality filtering is that the reference corpora are unlikely to contain examples of all data that are considered high quality. Thus, it is desirable to allow some stochasticity in the selection mechanism. For example, Brown et al. (2020a) use their classifier to score Common Crawl documents and keep each document only if they satisfy

$$\text{np.random.pareto}(\alpha) > 1 - \text{document\_score},$$

where $\alpha = 0.9$ was determined by matching the distribution of scores from the `WebText` corpus (Radford et al., 2019b). By selecting documents that don't match the "high quality" distribution with some stochasticity,

---

[10] https://commoncrawl.org/

they ensure that documents in the final corpus are mostly high-scoring, but still include some documents that are out of distribution with respect to the reference corpora.

When the utility metric is an importance weight (Xie et al., 2023b), the selection mechanism is to sample examples according to the importance weights across the dataset. Sampling (without replacement) according to the importance weights can be done efficiently with the Gumbel top-$k$ trick, which perturbs the unnormalized log-importance weights with Gumbel noise before selecting the top $k$.

**Potential challenges for quality filtering.**   An important consideration when performing quality filtering is that certain reference corpora may be biased towards or away from certain demographics, dialects, and socialects (Rae et al., 2021). Additionally, Gururangan et al. (2022) find that a quality filter trained on Wikipedia, books, and newswire (a replication of the quality filter from Brown et al. (2020a)), evaluated on high school newspaper articles, prefers articles from high schools in wealthier, higher-educated, and urban areas. They demonstrate that even though quality filtering is often assumed to be fairly neutral, it carries with it an implied judgment of preferred values. Thus, quality filtering should be done with care to ensure that biases are avoided and the distribution of data sufficiently covers the desired dialects and demographics. Filtering text for quality, while avoiding the introduction of biases is an important direction for future research.

It is still an open question whether using quality filters is a beneficial practice. Some recent works that have entirely foregone using a quality filter include `MassiveText` (Rae et al., 2021), `RefinedWeb` (Penedo et al., 2023), and `Dolma` (Soldaini et al., 2024). Specifically, Penedo et al. (2023) elect not to perform any quality filtering and to spend more compute on a stringent deduplication instead. It is not fully understood when quality filtering is beneficial and when it is not required, requiring further research.

---

**Data Quality Summary & Ideas**

- Data quality methods aim to select data points that are similar to data which is of known "high-quality".

- Classifier- and perplexity-based quality filtering are the existing approaches, with classifier-based filtering being the more popular method.

- For both classifier- and perplexity-based methods, it is important to be careful when training the classifier because this can introduce biases (*e.g.*, against lower-income, less-educated populations).

- 💡 Exactly which situations quality filtering is beneficial is still undecided, as recent works have trained performant models without using quality filters at all. This is an area that requires further study.

### 3.4 Domain-Specific Selection

While some language models are pretrained for general domain use, they can also be trained with a specific domain in mind, where selection can assist in finding data similar to the desired domain's distribution (*e.g.*, medicine or law). Domain-specific filtering methods mostly assume access to some in-domain data and additional data from auxiliary sources. The goal of filtering is then to find the auxiliary data which most resembles the distribution of in-domain data.

**Common utility functions.** In order to produce representations for data that can separate in-domain vs. out-of-domain data, utility functions for domain-specific selection often make use of models trained on one or both data distributions.

Many domain-specific selection methods stem from MOORE-LEWIS selection (Moore & Lewis, 2010) which follows from the following conditions. Let $I$ be an in-domain dataset, $N$ be a general purpose dataset, and $N_I$ be a subset of $N$ that is in-domain that we wish to discover. They note that the probability of a data point $x^{(i)}$ drawn randomly from $N$ being in $N_I$ is

$$P(N_I|x^{(i)}, N) = \frac{P(x^{(i)}|I)P(N_I|N)}{P(x^{(i)}|N)},$$

derived from a variant of Bayes rule. If the probability $P(N_I|x^{(i)}, N)$ could be directly measured, it would form a very strong utility function. The exact distributions are intractable, but estimates for $P(x^{(i)}|I)$ and $P(x^{(i)}|N)$ can be calculated by training language models on $I$ and a sample of $N$. Additionally, the probability $P(N_I|N)$ can simply be disregarded because it does not depend on $x^{(i)}$, so for all $x^{(i)}$ $\frac{P(x^{(i)}|I)}{P(x^{(i)}|N)} \propto \frac{P(x^{(i)}|I)P(N_I|N)}{P(x^{(i)}|N)}$.

In order to work directly with language model outputs, the ratio $\frac{P(x^{(i)}|I)}{P(x^{(i)}|N)}$ is converted into the log domain as $\log(P(x^{(i)}|I)) - \log(P(x^{(i)}|N))$. Finally, the cross-entropy loss from models trained on $I$ and $N$ is used to estimate each of these values.

More recent variants of MOORE-LEWIS selection have improved on the original idea. Axelrod (2017) propose *cynical data selection*, a theoretical framework for selecting data that maximizes the information gained by the model from each data point. Feng et al. (2022) empirically validate that the perplexity of encoder models improves when using cynical data selection compared to a "high-quality" subset (Wikipedia and Books). Xie et al. (2023b) reframe the ratio of probabilities as an importance weighting from an in-distribution and general purpose model.

Notably, while the original MOORE-LEWIS selection method suggests using a language model to estimate the utility function, modern language models have grown to sizes that would make the naive use of this method extremely costly. Thus, more efficient language models based on n-grams are typically used. For example, Xie et al. (2023b) utilize a bag of hashed n-gram model that can be computed very efficiently[11] and find that data selection with hashed n-gram features is strongly correlated with performance on target domains.

Engstrom et al. (2024) propose a model-dependent alternative to MOORE-LEWIS-based methods, which only depends on the data. They propose to use datamodels (Ilyas et al., 2022), a utility function that can approximately map a subset of training data to model performance after training on that subset. Generally, producing a datamodel requires training a series of models with different subsets of data. At very high filtering ratios (less than 25% of original dataset size), they demonstrate improved performance over DSIR (Xie et al., 2023b), but require significantly more compute.

In general, methods derived from MOORE-LEWIS selection have been the gold-standard, but it may be possible to adopt other methods of data attribution and valuation, similar to datamodels, for the purpose of selecting in-domain data. However, these methods typically have higher computational requirements, a challenge that needs to be addressed. Perplexity-based methods (similar to perplexity-based quality filtering methods (§3.3)) that only require an in-domain model are also a plausible research direction.

---

[11]https://github.com/p-lambda/dsir

**Selection mechanisms and thresholds.** Domain-specific filtering methods can utilize either deterministic or stochastic selection mechanisms. Most MOORE-LEWIS-based methods tend to use deterministic selection with a threshold. As previously mentioned, domain-specific filtering methods often assume access to in-domain data in addition to the auxiliary data. In order to select an appropriate threshold sensitivity, a held-out set of in-domain data can be used (Sethy et al., 2006). It is also possible to use stochastic selection mechanisms. For example, DSIR (Xie et al., 2023b) resamples the data according to the importance weights. A stochastic mechanism can allow for a wider diversity of data, especially when there are minority subpopulations of the target that may be ignored when deterministically selecting high-scoring data under some metric.

**Comparison to data quality filters.** Domain-specific filtering methods are quite similar to data quality filtering and can be viewed as a generalization of data quality filtering. The core difference is that in data quality filtering, the in-domain data (reference corpus) is used as a proxy for the desired distribution but in reality this is not guaranteed, thus an element of stochasticity is crucial for data quality filtering. On the other hand, given enough in-domain data, domain-specific filters can directly measure whether auxiliary data matches the desired distribution.

**Challenges for domain-specific selection.** Prior methods of domain-specific selection utilize just a model of the in-domain data, but these methods can skew the resultant distribution towards the modes of the existing in-domain data (Sethy et al., 2006). For example, it is possible that the in-domain data contains some very common phrases (*e.g.*, "okay") which are not necessarily representative of the domain. To avoid this issue, it is possible to perform deduplication on the in-domain data. However, a delicate balance must be reached, as deduplication can skew the in-domain data away from valid modes. Domain-specific selection methods that utilize a general-domain model in addition to the in-domain model (*e.g.*, MOORE-LEWIS selection) handle this issue quite well and thus it is highly recommended to use such methods.

---

**Domain-Specific Selection Summary & Ideas**

- Domain-specific selection aims to find data most similar to a specific domain (*e.g.*, medicine or law).
- MOORE-LEWIS style methods (requiring one in-domain model and one general purpose model) have stood the test of time and are still the most popular methods for domain-specific filtering.
- 💡 Datamodels and other methods based on data attribution and valuation can be explored as competitive methods, but generally have higher computational requirements.
- 💡 Perplexity-based methods may be another direction to explore. They require only an in-domain model, which could improve efficiency. It's possible that they would not perform as well as MOORE-LEWIS based methods, but because they use less compute they could be used in multiple rounds of selection, as in bootstrapping.

---

### 3.5 Data Deduplication

Datasets originating from internet dumps often abound with duplicate and near duplicate documents (Elazar et al., 2024; Magnusson et al., 2023). A dataset with duplicated data, or near duplicates, increases the distribution density around those areas. In cases where these high-density points are of high value, keeping near duplicates can be beneficial. However, for pretraining where the exact evaluation distribution is unknown, it is generally preferred to remove duplicates so that the training distribution provides greater coverage with less redundancy. Carlini et al. (2023) and Kandpal et al. (2022) demonstrate a direct relationship between the number of times a data point is duplicated and the degree to which a model will memorize the data point, a relation which increases with model scale. Additionally, filtering out duplicates can reduce the training time of machine learning models (Sorscher et al., 2022) and has been shown to sometimes improve accuracy on downstream tasks (Lee et al., 2022a; Tirumala et al., 2024).

**Common utility functions.** Deduplication can come in multiple forms, where most approaches are based on **URLs, hashing, string metrics**, and **model representations**. Some methods find exactly matching strings and documents while others use approximate matching methods calculated according to some similarity measure.

**Exact match utility functions.** Determining *exact matches over whole documents* is fairly straightforward, and can be performed cheaply. **URL deduplication** is a common first step when using a snapshot of crawled web pages. It is possible for individual web pages to appear multiple times in the snapshot, so removing data that shares the same URL is a very cheap and efficient method for initial deduplication (Agarwal et al., 2009; Soldaini et al., 2024).

Some exact-matching utility functions utilize hash functions. These **hash-based** methods are guaranteed to find all exact matches but, due to the possibility of collisions, may also find false positives (accidentally removing non-matching documents), although Elazar et al. (2024) demonstrate that a simple hash function on large scale datasets can experience no collisions. CCNet (Wenzek et al., 2020) performs deduplication by first normalizing all text, then computing a hash code for each document, and finally comparing the first 64-bits of SHA-1 digits of the hashed documents. When developing `OSCAR`, Ortiz Suárez et al. (2019) use a simple non-collision resistant hashing algorithm to determine whether whole documents are exact matches. Another common method for deduplication is the Bloom filter which utilizes a space-efficient bit array to compare hashed documents (Bloom, 1970; Soldaini et al., 2024).

Determining *exactly matching text segments*, as opposed to whole documents, is more expensive, but equally as important to remove. Lee et al. (2022a) propose the ExactSubstr algorithm that finds examples which share sufficiently long substrings. In their case, their utility function determines that a pair of examples are duplicates if they share a 50-token substring. To perform the deduplication efficiently, they utilize suffix arrays (Manber & Myers, 1993) which allows them to find duplicates in linear time. This method raises the question: how much text surrounding the duplicate should be removed? Lee et al. (2022a) remove only the exact substring that is duplicated, while Penedo et al. (2023) experiment with dropping the entire document, or masking the loss of the duplicated span, finding that all methods lead to similar downstream performance. Similar to the use of Bloom filters for matching entire documents, they can also be used to detect matching segments of any size.

**Approximate match utility functions.** Removing approximately matching documents is generally more expensive to compute than exact matches, and is often performed as a secondary step after exact match deduplication (Laurençon et al., 2022).

Similarly to exact matching methods, **hash-based** methods are very commonly used for approximate matching methods. For example, MinHash (Broder, 1997) is an approximate hashing algorithm that excels at finding templated documents (*e.g.*, licenses with only specific entities differing or placeholder SEO text repeated across websites). Modern deduplication methods use other variants of hashing such as MinHashLSH (Brown et al., 2020a) and SimHash (Charikar, 2002) that utilize locality sensitive hashing.[12] The general method

---

[12]See the blogpost by Mou (2023) for implementation details on MinHashLSH.

behind hashing methods for deduplication is to first group small segments of each documents (often called shingling), then to encode each segment using a large number of hashing functions into features. Next, aggregate the features (*e.g.*, MINHASHLSH keeps only the smallest features). Finally, documents can be compared with some method of similarity computation between features (*e.g.*, MINHASHLSH uses a Jaccard similarity) and removing documents below some threshold. When creating `MassiveText`, Rae et al. (2021) use MINHASH with 13-grams and 450 hash functions and set the jaccard similarity threshold at 0.8, randomly removing either of the documents that are determined to be duplicates. In contrast, Gao et al. (2020) use only 10 hash functions, but Rae et al. (2021) find that the more aggressive deduplication leads to better model performance.

In addition to hashing techniques, **string metric-based** methods can be used for futher filtering, although they are generally more computationally expensive. For example, Lee et al. (2022a) compute an edit similarity metric based on token sequences using their token edit distance. Furthermore, when filtering code files, Chowdhery et al. (2023) find duplicated files based on a Levenshtein distance.

Finally, **model-based** approximate matching techniques utilize a learned representation of data points to deduplicate similar data. These methods can be very costly, but allow for more flexibility by considering the semantics of text. For example, SEMDEDUP (Abbas et al., 2023) uses a pretrained 125M OPT model and calculates a representation for each data point in the `C4` dataset by using the last layer embedding of the final token in the document. Then, they cluster data points to create groups of rough similarity, followed by computing the pairwise cosine similarity of data points within each cluster. Document De-duplication and Diversification (D4) (Tirumala et al., 2024) go even further, using a pipeline of first deduplicating documents with MINHASH, then applying SEMDEDUP (Abbas et al., 2023) and clustering the remaining data using K-means, and finally applying the SSL prototypes method of Sorscher et al. (2022) which removes the "most prototypical" examples in each cluster. The authors demonstrate that deduplication with multiple methods leads to improved performance as each method is designed to remove only some aspect of similarity. One concern with these methods is that they have only been tested on CommonCrawl data, which is known to be quite noisy, and have not been demonstrated to be beneficial on cleaner datasets. Prior work has shown that the embedding performance of pretrained decoder language models such as OPT 125M is suboptimal (Muennighoff, 2022). Thus, a promising future direction for this line of work is using better embedding models that are specifically trained for clustering (Muennighoff et al., 2022a; 2024; Xiao et al., 2023).

**Challenges for data deduplication.**  One concern with hashing-based method for approximate deduplication is that because hashing functions do not consider the order of each individual shingle (the chunks of text used to create the hashed representation), they act similarly to bag-of-word models where long documents are more likely to have similar representations. Laurençon et al. (2022) found that this led to long documents having a higher percentage of false positive as duplicates. Thus, they did not remove documents in a cluster of near-duplicates if a document is longer than 6000 characters. Instead, they use substring deduplication based on the suffix array (as in Lee et al. (2022a)) for documents with more than 6000 characters.

Methods can search for duplicated documents, or just chunks of text which are duplicated within and across documents. Once duplicated chunks have been found, the question arises: what to do with these chunks? The EXACTSUBSTR method Lee et al. (2022a) removes only the exact substring that is duplicated and Penedo et al. (2023) also elect to remove only the duplicated span and keep the rest of the document. This can lead to incoherent documents, but Penedo et al. (2023) empirically find it to work better than removing the entire document. Another option is to loss-mask the duplicated tokens (prevent the loss from being computed on duplicated text), but this adds code complexity as the start and end of each duplicated span will need to be tracked and considered when training the model.

---

**Data Deduplication Summary & Ideas**

- Deduplication is a very important step and is often used multiple times within the data selection pipeline: first with a simple URL-based filter, then with more complicated hashing- and model-based methods.
- Deduplication generally relies one of four methods: URLs, hashing, string metrics, or model representations.
- Bloom filters are often used as state-of-the-art exact deduplication (even though they can remove non-duplicated data) as they are very space efficient.
- For a more thorough deduplication, approximate matching methods can be used, where MinHashLSH is currently the most commonly used method.
- 💡 In settings where data efficiency is the primary goal, model-based deduplication has demonstrated good results. However, these methods are relatively new and need further study to demonstrate their benefits in other cases (*e.g.*, when performance is the primary goal).

---

### 3.6 Filtering Toxic and Explicit Content

Practitioners often aim to curate a dataset that is "unbiased and fair". As such, the aim is for models trained on such data to follow similarly unbiased behavior. Certain popular datasets have also been found to have illegal or extremely inappropriate content, which motivates strict data filters (David, 2023). Towards this purpose, most datasets employ a process to filter out documents that do not conform with such behaviors. Bias and fairness are one of the least well-defined filtering requirements and may be in conflict with different ideologies (Gururangan et al., 2022) and implications (Longpre et al., 2023c). A crucial aspect of this topic is the lack of transparency (Bommasani et al., 2023), requirements, and the downstream effects of applying bespoke bias filtering to pretraining data. Many works solely provide minimal information about the heuristics used for filtering out "biased" data without reporting any ablation on the impact on downstream tasks. A notable exception is the work of Longpre et al. (2023c), which controls and ablates for some of these filters to comprehensively study their effects. They find that using toxicity filters (specifically, Jigsaw's Perspective API[13]) leads to models that produce less toxic texts, however, the model's ability to detect toxic content also decreases. As such, one must be careful when using such filters depending on the end-application.

In what follows we summarize the filters related to bias and fairness used in different papers based on the filter types. As mentioned above, such filters are often based on heuristics and the effect of such filtering is not well studied.

**URL-based methods.** Simple filters that aim to remove NSFW content can filter solely based on the source URL, a piece of metadata that is typically stored alongside the text of datasets. Datasets such as RefinedWeb (Penedo et al., 2023) filter out documents coming from URLs that match words from a curated list, as well as from blacklisted domains.

**Lexicon-based methods.** One of the most common methods simply filters documents containing words or phrases that match a lexicon. It is often unclear what an appropriate threshold is for this strategy (*e.g.*, whether a single match, or multiple matches, are required to justify the exclusion of an entire document), but many datasets use such methods (Raffel et al., 2020; Xue et al., 2021; Achiam et al., 2023). Laurençon et al. (2022) go further and adapt such lexicon based on the detected language, to tailor such toxicity filters across multiple languages.

**Classifier-based methods.** Another common approach is to score documents using a classifier that aims to detect "toxic" documents. Such classifiers are often trained using a simple classifier (*e.g.*, a linear model over bag-of-words embeddings) so it can be applied at scale on a large dataset. Others use an external,

---

[13]https://www.perspectiveapi.com/

blackbox tools for classifying toxicity, *e.g.*, Jigsaw, or Google's SafeSearch[14] (Rae et al., 2021; Longpre et al., 2023c).

**Perplexity-based methods.** A less common method is to score documents using a model trained only on the undesirable content. Jansen et al. (2022) train a model directly on adult and harmful content and then use the model's perplexity as a utility function where documents with a high perplexity are assumed to not contain any adult or harmful content.

**Personally Identifiable Information obfuscation.** Finally, Personally Identifiable Information (PII) is another source of concern, which may expose the personal information of individuals. Frequent targeted PIIs are phone numbers, email addresses, IP addresses, and secret keys. A common approach for handling PII is to detect them using regex (Subramani et al., 2023; Elazar et al., 2024; Soldaini et al., 2024; Groeneveld et al., 2024) or classifiers (Allal et al., 2023). Once PII terms are detected, a common practice is to obfuscate them with special tokens.

---

**Toxic and Explicit Content Filtering Summary & Ideas**

- There is a wide variety of toxic and explicit content that are frequently considered undesirable to have in training or evaluation data.

- Common methods for filtering out toxic and explicit content include blacklisting certain web domains, blacklisting certain words, using a classifier to detect harmful content, using perplexity to detect harmful content, and removing PII with regular expressions or classifiers.

- 💡 There have been limited studies on the effects of these filters, warranting further research on the impacts of such filtering to understand how they affect the distribution of data and the downstream models behaviors.

- 💡 A useful direction for research is how to mitigate any negative impacts of this filtering, or conversely, how can toxic and explicit behaviors be reduced at other stages of model development (*e.g.*, *alignment*), or possibly even at evaluation time through decoding methods.

---

[14]https://support.google.com/websearch/answer/510

### 3.7 Specialized Selection for Multilingual Models

Besides filtering for the desired languages as detailed in Section 3.1, there are additional considerations when dealing with multilingual pretraining data.

**Language-specific filtering hyperparameters.** Many selection methods, such as deduplication, can be reused across languages (Laurençon et al., 2022). However, it is critical to adapt the hyperparameters for the respective language and often manually tune them (Laurençon et al., 2022; Scao, 2023; Scao et al., 2022; Workshop et al., 2022). For example, while heuristics based on technical characters (`[0-9{}+/()>]`) are still valid, for a length-based filter (discussed in §3.2) one needs to consider that languages like Chinese have much higher information density per character leading to the same content requiring fewer characters than English. Thus, in a filtering pipeline Chinese would require a smaller cut-off value for minimum text length than English (Scao, 2023). Additionally, when using language identification models (*e.g.*, FASTTEXT) to detect the language of a document, it is important that the threshold for removal is determined by native speakers, and specific to each language (Laurençon et al., 2022). Not only is the language ID used for determining the language of a document, but it can also be used at the sentence level, where Kudugunta et al. (2023) consider removing sentences if the sentence level language ID and document level ID do not match.

**Script-based filtering.** Languages with non-Latin scripts (*e.g.*, Mandarin, Arabic, and Hindi) commonly suffer from incorrect encoding which may require additional filtering or noise correction (Kudugunta et al., 2023). Further, some scripts such as Dongba symbols are not yet supported by Unicode, making the selection of such data especially difficult. Using script-based filters can also serve as language identification for languages that have a one-to-one mapping with a script, such as Japanese Hiragana (§3.1).

**Manual selection.** Due to difficulties in language identification and data availability, noisy language classifications tend to correlate with language rarity (Kudugunta et al., 2023). Thus, it is often necessary to manually inspect multilingual datasets to ensure language identification is correct and data in low-resource languages is useful (Kudugunta et al., 2023). Kreutzer et al. (2022) demonstrate that simply ignoring this issue has led to at least 15 corpora with no usable text, while a significant portion of the available multilingual datasets contain less than 50% of acceptable quality. While large-scale collaborations on projects such as `ROOTS` (Laurençon et al., 2022) have been able to outsource this to speakers of each language, this filtering step can also be effective when done by people who do not speak the respective language. Using translation services to gauge the quality and looking for irregularities is possible even for non-speakers (Kudugunta et al., 2023).

---

**Specialized Selection for Multilingual Models Summary & Ideas**

- Many of the selection methods from other sections in this work can simply be reused when filtering for multilingual data (*e.g.*, deduplication, heuristics on technical characters, data mixing), but some require special parameters for each language (*e.g.*, length-based heuristics).

- 💡 Particularly for low-resource languages, native speakers must be incorporated into the filtering process to ensure that the data maintains an acceptable quality. Aya (Singh et al., 2024) is a successful example of how this type of inclusion and collaboration can look.

### 3.8 Data Mixing

Large pretraining corpora often comprise a mixture of data from $k$ domains, where domains are typically defined by provenance. The weighting across domains are specified by the domain weights $\alpha \in \Delta^k$, which comprise a $k$-dimensional probability vector. For example, the `Pile` dataset (Gao et al., 2020) is composed of 24% web data, 9% Wikipedia, 4% GitHub, etc.[15] The choice of domain weights can result in significant differences in downstream accuracy (Xie et al., 2023a; Albalak et al., 2023a; Xia et al., 2024a; Fan et al., 2023). Given a fixed training budget, data mixing methods optimize the domain weights to improve training efficiency and model performance.

**Utility function and selection mechanism.** Data mixing methods work with a constrained utility function and selection mechanism. We define a domain as a set of data points $D \subseteq \mathbb{D}$. Given a set of domains $D_1, \ldots, D_k$, in all data mixing methods, the utility value $v(x)$ for a data point $x \in D_i$ is equal to the normalized domain weight $\alpha_i/|D_i|$. In particular, the utility value $v(x)$ is constrained to be equal to $v(y)$ whenever $x, y$ are elements of the same domain. Consequently, any two domains with a nonempty intersection would have the same utility, and thus data mixing methods typically work with domains that are mutually disjoint. The selection mechanism simply samples data points according to the utility function (which is a valid distribution over data points) in a hierarchical manner. In particular, to sample a data point, we first sample a domain index $I \sim \text{Categorical}(\alpha)$ and then sample a data point uniformly within that domain $D_I$.

**Common baselines.** Using **heuristic or manually determined** weights is a common baseline. Defining the domain weights according to the natural domain sizes weights all the individual data points equally (Raffel et al., 2020). Beyond this, popular datasets and models (*e.g.*, the `Pile` (Gao et al., 2020), LLAMA (Touvron et al., 2023a)) often use heuristics to adjust the data mixture, upweighting domains with intuitively higher quality text that have likely gone through an editing process, such as books and Wikipedia articles. However, these manually-determined domain weights are likely not optimal (Xie et al., 2023a; Albalak et al., 2023a).

**Empirically determined** weights provide another common method, where the domain weights are tuned according to the performance of a model on downstream tasks. This approach tends to be expensive since it may require pretraining many models within a zeroth-order optimization algorithm or heuristic search strategy. For example, the `GLaM` dataset (Du et al., 2022) sets mixture weights based on (1) the downstream performance of smaller models and (2) the sizes of each domain, preventing small sources such as Wikipedia from being over-sampled. While the size of these smaller models is unknown, these models must be large enough to have a meaningful downstream performance (roughly 1B parameters). Similarly, the dataset used to train GOPHER (Rae et al., 2021) used domain weights tuned by sweeping over 7 combinations of domain weights and choosing the one that leads to the best downstream performance.

Overall, the optimal domain weights may not be intuitive and likely depend on all parts of the training pipeline, including the tokenizer, dataset, model architecture, optimizer, and other hyperparameters. For example, recent work has shown that repeating data can provide similar benefits to adding new data, up to a limit (Muennighoff et al., 2023b). Muennighoff et al. (2023b) also find that including a large proportion of code (*e.g.*, 50%) does not harm natural language performance, while having the additional benefit of improving performance on reasoning-based tasks.

**Principled approaches.** Recent works have derived their data mixing methods based on well-studied fields and concepts including information theory, importance sampling, distributionally robust optimization, and multi-armed bandits. We divide these methods into two categories: methods that determine the weights separate from model training (offline), and those which determine the weights during model training (online).

**Offline data mixing** methods optimize a static set of domain weights, which define a new reweighted dataset for training a language model. DOREMI (Xie et al., 2023a) optimizes domain weights without using downstream tasks by aiming to achieve a low loss on all domains, formalizing this problem as group distributionally robust optimization (GROUP DRO (Oren et al., 2019; Sagawa et al., 2020)). To avoid the pessimism of DRO, DOREMI optimizes the excess loss of each domain with respect to a pretrained reference

---

[15] The mixture weights depend on the tokenizer.

model. DoReMi first optimizes the domain weights with a small proxy model, then uses the optimized domain weights (the time-averaged domain weights over the optimization trajectory) to train a larger model. DoReMi uses a multiplicative update by an exponential function of the excess loss, which is derived from online mirror descent and is equivalent to the exponentiated gradient/Hedge algorithm update. Similarly, DoGE (Fan et al., 2023) optimizes the domain weights with a two-stage process that generalizes across model scales, with the same update rule. However, instead of optimizing the excess loss, DoGE optimizes the gradient alignment between each domain and all other domains, with the goal of putting higher weight on domains with better generalization or transfer to other domains.

**Online data mixing** methods change the domain weights for sampling examples during training, allowing for the algorithm to enforce an ordering on the training data. Skill-it (Chen et al., 2023c) uses a similar multiplicative weights update to the offline methods, but additionally scales the losses by a matrix that captures pairwise interactions between domains. This matrix is learned from data by training many models on pairs of domains. Skill-it considers both domains defined by task or "skill" as well as by provenance. ShearedLLaMA (Xia et al., 2024a) uses an online variant of DoReMi (Xie et al., 2023a) that replaces the pretrained reference model with estimates of the reference model loss, computed using domain-level scaling laws fitted on the Pythia (Biderman et al., 2023b) suite of pretrained models. ODM (Albalak et al., 2023a) proposes an efficient online data mixing method based on EXP3 (Auer et al., 2002), an extension of the Hedge algorithm in the adversarial bandit setting. Motivated by the goal of maximizing information gain during model training, ODM formulates the reward for each domain as the loss/perplexity of each domain (directly proportional to information gain). In the machine translation setting, Schioppa et al. (2023) aim to train a model using a mix of standard language modeling data and parallel machine translation data. They propose an online data mixing method that optimizes a relative loss improvement score, and updates the domain weights by simply keeping a moving average of the scores.

**Challenges of data mixing.** There is a natural trade-off in data mixing; when increasing the proportion of data from one domain the distribution density increases for that domain, but the relative proportion (density) of all other domains decreases, which can lead to poorer performance on less represented domains.

One challenge that has not been addressed in any of these works is how their weights transfer to new settings. For example, Albalak et al. (2023a) find that the weights determined by Xie et al. (2023a) do not transfer well when using a different tokenizer. It is unclear how to transfer the weights learned under one tokenizer to a new tokenizer without learning the weights from scratch. Furthermore, none of the principled approaches discuss the effects of extrapolating the calculated weights to training on larger quantities of tokens. To exemplify the problem, consider that the weights are calculated by training a model on 50B tokens, and the weights are then used to train a model on 1T tokens. Any domains that saw 1 epoch at 50B tokens will now see 20 epochs at 1T, leading to possible overfitting on smaller domains.

---

**Data Mixing Summary & Ideas**

- When a pretraining corpus is composed of data from multiple domains, data mixing is an important step of data selection that assigns weights to each domain, determining the final composition of the dataset.

- Weights can be determined manually, by heuristics, empirically determined, or from principled approaches. Each of these methods have been proven under specific evaluations, but none has proven as a general-purpose best method.

- 💡 Data mixing is challenging, as there is a natural trade-off when upweighting one domain, the remaining domains must be downweighted. Research on overcoming this trade-off may be difficult, but would be very impactful. One possible direction are methods that consider the interactions (similarities) between domains, which could identify domains that can be downweighted without reducing performance, and those specific domains whose weights cannot be reduced without detriment to performance.

- 💡 Each of the approaches discussed has different benefits, and combining approaches (*e.g.*, 1/3 heuristic, 1/3 empirically-drive, 1/3 principled) could lead to better general purpose use, but how to best combine them leads to additional complexity.

- 💡 Although principled approaches are prominent and have lead to state-of-the-art mixing, the weights they determine can also be brittle when used under different settings (*e.g.*, tokenizer, longer training), which are issues that need to be addressed and explored in future research.

---

### 3.9 Current Best Practices and Landscape For Data Selection

The previous sections have described, in detail, a wide variety of works, methods, and datasets that are relevant to the data selection process. Data selection is an open and active area of research and while the exact details of best practices may change over time, there are some practices and ideas that are of high importance and will likely endure the test of time. In this section we take a step back to look at the high-level of the current landscape of research on data selection, and best practices as they currently stand.

#### 3.9.1 Best Practices

Best practices are actively being developed, and new models are consistently being released with new claims of being state-of-the-art. While there are no blanket statements of what is "best" in all use-cases for language models (because best can only be in reference to a specific evaluation protocol or benchmark, not an actual capability), we describe some trends that have either stood the test of time or have been adopted rapidly and widely. The first trend to notice is that filtering is generally performed starting with the methods that are most efficient and remove broad quantities data, with the later filters being those which are more specialized, but often more costly.

**Language filtering.** When filtering large quantities of data, an important first step is to ensure that the data is from the desired language, where the most commonly used method is the fastText classifier which has been trained to recognize 157 languages and runs very efficiently.

**Heuristic filtering.** The next step is usually to remove data using heuristics that are very fast to compute. Examples include removing documents with fewer than 50 or greater than 10,000 words or with fewer than 3 sentences. Examples of commonly used heuristics are those from C4 (Raffel et al., 2020) and MassiveText (Rae et al., 2021). These heuristics can be accessed through the Dolma toolkit (Soldaini et al., 2024) or DataTrove (Penedo et al., 2024).

**Data quality.** Filtering for "high-quality" data (data which was written by humans and has likely gone through an editing process) is a useful step when processing large quantities of web data, but less likely to be useful for data collected from specific domains (*e.g.*, scientific articles or legal documents). The most common method for quality filtering is to train a classifier that can differentiate between random web data and data from a known high-quality domain. There are no commonly used open-source models, but the methods have

been described within previous works (Brown et al., 2020b). A common approach when using quality filters is to use a stochastic filter, which will allow some data with a low quality score to pass through. This can help to reduce some of the biases known to exist in quality filters.

**Domain-specific filtering.** Methods for domain-specific selection are quite similar to the methods for data quality, but only required when training a model for a specific target domain. The goal of these methods will be to find data within a large general purpose corpus which is most similar to a target domain. The current best methods, such as DSIR (Xie et al., 2023b), train two models: one in-domain and one general domain model. To determine whether a data point is similar to the target domain, the difference of probabilities under each model is calculated.

**Deduplication.** Deduplication has proven to be a beneficial step in every pretraining data selection pipeline. There are multiple methods of deduplication that vary from very simple and broad deduplication methods which only consider metadata to very fine-grained deduplication on individual sentences. When using web-text, a simple and cheap first step is to deduplicate data based on URLs. Next, hash-based methods such as MinHash (Broder, 1997) (approximate matching) and Bloom filters (exact matching) have been used with great success. The previously mentioned methods are fairly cheap, but only consider word or token matching, and do not consider semantic matching. Model-based methods such as SemDeDup (Abbas et al., 2023) have demonstrated success for deduplicating semantically similar data, but are much more computationally expensive and should generally be applied later in the filtering pipeline after cheaper filtering methods.

**Filtering toxic and explicit content.** Removing toxic and explicit content from web text is generally performed through multiple steps to handle different types of content. NSFW content can be removed, in part, by filtering out URLs that match NSFW word lists and from blacklisted domains. Beyond just searching for bad words in the URL, it is also common to filter out documents that contain the same list of NSFW words. Removing personally identifiable information (PII) is another very important step and is commonly found and removed using regex filters, a good example of which is Dolma (Soldaini et al., 2024).

**Selection for multilingual models.** Generally speaking, most methods from the other steps will work the same for multilingual models, with the exception that some hyperparameters may need to be adjusted to work for each language. An important best practice for creating multilingual models is to incorporate native speakers into the decision making process to ensure that data maintains an acceptable quality.

**Data mixing.** Data mixing is only used when the training dataset is composed of data from multiple clearly defined domains or sources (*e.g.*, books or web text). As of writing, there are no clear best practices for automated methods for data mixing as this is still a very young area of research. Methods of determining domain weights automatically tend to use loss-per-domain as part of a signal as to the importance/difficulty of each domain. Starting points for research are DoReMi (Xie et al., 2023a) and ODM (Albalak et al., 2023a).

### 3.9.2 Current Landscape of Datasets For Data Selection Research

The most popular datasets used for data selection until now have been RedPajama (Skill-it (Chen et al., 2023c), SlimPajama (Shen et al., 2024), data mixing laws (Ye et al., 2024)), The Pile (ODM (Albalak et al., 2023a), DSIR (Xie et al., 2023b), DoReMi (Xie et al., 2023a), Selection for encoders (Feng et al., 2022)), and C4 (DsDm (Engstrom et al., 2024)), each with pros and cons. C4 (Raffel et al., 2020) (750Gb of data) is the oldest and contains only webcrawled data. C4 generally leads to models with lower benchmark scores as the other 2 datasets, but this makes it an easy first testing ground for data selection methods since there is increased room for improvement. The Pile (Gao et al., 2020) (800Gb of data) dataset contains slightly more data than C4, but only ~28% of the total dataset comes from web scrapes and the remaining 72% comes from a wide variety of domains including books, scientific articles, and Github. This combination of well-defined sources makes the Pile a good candidate for research on data mixing. Lastly, RedPajama (1T tokens) is the newest and the largest of the commonly used datasets, and was originally intended to be an open-source re-creation of the dataset used to train Llama-2. Recently, RedPajama-2 (30T tokens) was released which may be an even better resource for testing data selection. RedPajama-2 comes with precomputed quality signals including those implemented for C4 and RefinedWeb, allowing for experimentation on selection mechanisms under controlled settings. Another beneficial aspect of RedPajama-2 is that it contains a huge amount of data, allowing researchers to filter the data down to various scales, ranging from billions to trillions of tokens.

## 4   Data Selection for Instruction-Tuning and Multitask Training

Instruction-tuning and multitask training are methods for addressing the mismatch between the pretraining data distribution and downstream use cases. Multitask training is a method where the model is trained on a wide variety of supervised tasks with the goal of performing all the training tasks and possibly generalizing to unseen tasks. Recently, instruction-tuning has become a dominant training paradigm where the model is trained on pairs of (`Instruction`, `Output`), where `Instruction` denotes a human instruction for the model, and `Output` is the desired output, or an example of a desirable output. The goal of instruction tuning is to encourage the model to generate outputs in a way that is more controllable and helpful for the user (Zhang et al., 2023c).

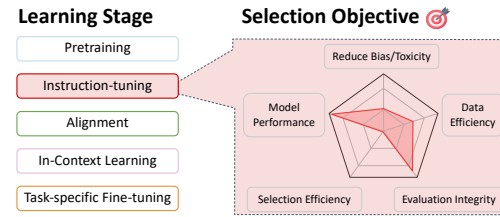

Figure 5: **Data selection: instruction-tuning.** Generally, the most important objectives when selecting data for instruction-tuning are *model performance* and *evaluation integrity.*

Both multitask training and instruction-tuning generally assume that the model has been pretrained and has already acquired fundamental language capabilities. The intent of models that have been trained with either multitask training or instruction-tuning is, broadly speaking, to handle a very wide variety of possible inputs for downstream use cases as either classification or generative models. Thus, the aim of data selection for these settings is typically focused on collecting a wider variety of data, and diversifying the data that already exists.

**Diversification by scaling tasks and datasets.**   As model sizes have scaled up and model capacity has correspondingly improved, distribution diversification has proven to be an effective method of improving model generalization, for both specialized tasks and general purpose model performance (Wei et al., 2021). Methods of multitask training (*e.g.*, MT-DNN (Liu et al., 2019), Muppet (Aghajanyan et al., 2021), T5 (Raffel et al., 2020), ExT5 (Aribandi et al., 2021)) demonstrated the benefits of training on a range of datasets and tasks, selected for their perceived high-quality curation. Subsequently, a number of works (Khashabi et al., 2020; McCann et al., 2018; Keskar et al., 2019) proposed unifying the format of any language task into questions and answers, effectively adjusting the data point characteristics, with the goal of better matching the train and test distributions and expanding the number of tasks a model could learn in tandem, further improving model generalization. Based on this premise, `Natural Instructions` (Mishra et al., 2021), `FLAN` 2021 (Wei et al., 2021), and `P3` (Sanh et al., 2022) scaled up the number and diversity of training tasks, distinguished by templated instructions—now known as *instruction tuning.*

At this stage, data selection was based mainly on availability across open-sourced dataset repositories (*e.g.*, Hugging Face and TensorFlow Datasets), and pressure to scale, rather than any principled selection criteria. To extend this scaling beyond the openly available data, data augmentation techniques such as translation (Muennighoff et al., 2022b; Yong et al., 2022; Dhole et al., 2021; Singh et al., 2024; Üstün et al., 2024), input inversion (Min et al., 2022; Longpre et al., 2023a), and negative examples (Wang et al., 2022b) have successfully been used to further enrich the data variety. A number of works also increase data diversity by templatizing data in multiple prompt formats, including zero-shot, few-shot, and chain-of-thought prompts (Chung et al., 2022; Iyer et al., 2022). These works demonstrate that task variety and template variety both drive greater generalization, though Wei et al. (2021) report easier training tasks, such as sentiment analysis, contribute less.

**Manual and heuristic-based diversification.**   While scaling tasks has been an effective method of diversifying data, naively including all data possible can lead to severely imbalanced datasets and has motivated the use of better mixing and balancing techniques. For example, Wei et al. (2021) impose a maximum number of sub-dataset examples to prevent large datasets overwhelming the training mix. Additionally, Iyer et al. (2022) experiment with a range of maximum values, finding that including at most 512 data points from each sub-dataset achieves the best results across held-out test sets. By setting an upper limit to the number of samples from any individual sub-dataset, these methods improve the data diversity

by reducing the distribution density around arbitrary regions of the sample space, as defined by individual sub-datasets.

Iyer et al. (2022); Longpre et al. (2023a); Chung et al. (2022); Wang et al. (2024) achieve strong performance improvements by varying the proportions of sub-mixtures with a manual grid search, but this can be very costly. Furthermore, Wang et al. (2024) and Ivison et al. (2023b) show that none of the available mixtures are by default best on all evaluations. This strongly motivates the need to design and explore data selection methods in the instruction-tuning setting.

**Model-based diversification.** More recently, instruction-tuning data has been further diversified through the use of existing models using conversation as the unified data format. Training data can be generated directly by a model, often from the OpenAI API, creating and expanding examples iteratively (Honovich et al., 2022; Wang et al., 2023c; Xu et al., 2023; Taori et al., 2023; Lee et al., 2023; Kim et al., 2023). Wang et al. (2023c) begin with a small seed pool of example tasks, which the model uses to iteratively generate new tasks and instances of that task, followed by filters to ensure that data is "high-quality" and to remove duplicated data. EVOL-INSTRUCT methods enhance these techniques to iteratively re-write training instances to be more complex and challenging (Xu et al., 2023). Lu et al. (2023a) iteratively trains on and produces its own training data with self-refinement techniques. These methods are shown to produce a diverse array of prompts that can be guided to match the distribution of certain real world tasks, domains, and topics which are not covered in existing datasets such as P3, FLAN, or the dataset used to train OPT-IML.

A survey of synthetic data tasks showed that they tend to focus on longer-form responses than what exists in academic datasets, as well as tasks related to brainstorming, planning, explanation, and creativity (Longpre et al., 2023b). The advent of synthetic data has also spurred developers to filter out system-specific outputs, such as disclaimers and refusals (*e.g.*, "I'm sorry, but[...]"[16] from OpenAI models).

**Data efficiency.** As larger and more diverse instruction tuning collections have become available, the benefits of scale seem to have diminished, and in response recent methods have explored a variety of utility functions aimed at improving data efficiency while focusing on quality, diversity, difficulty, and complexity as components of the utility. LIMA demonstrated strong results with only 1000 highly curated, diverse training instances meant to approximate real user prompts with high-quality responses (Zhou et al., 2023). Similarly, the best performing variants of Open Assistant are high-quality subsets, as graded by human reviewers (Köpf et al., 2023; Muennighoff et al., 2023a; Zhuo et al., 2024). The previous works use human judgments to determine quality, but new work has developed methods to automatically measure and sub-select for instruction data quality and diversity. For instance, Cao et al. (2023) use natural language indicators (such as input/output lengths, perplexity, and textual lexical diversity) with BlendSearch to measure and identify the highest quality data points. Similarly, InsTag (Lu et al., 2023b) labels instructions based on semantics and intentions. On the other hand, Lian et al. (2023) improve data efficiency through the use of a model-based quality filtering. They perform a pass over the FLAN dataset and remove datapoints where the answer appears to be incorrect (according to GPT-4), then train a model on this reduced dataset and find they get a model with similar quality level, but only 2/3 the compute. Similarly, Chen et al. (2023b) use ChatGPT to score data points and use this score utility function to select a subset of data, improving data efficiency compared to using the full Alpaca (Taori et al., 2023) dataset. Additionally, Li et al. (2023a) introduce the *Instruction-Following Difficulty* metric which allows a model to self-select the data points which are most important. Data Efficient Instruction Tuning for Alignment (DEITA) surveys InsTag and other prior methods that measure data complexity, quality, and diversity, and propose a unified data selection strategy that samples high quality examples, while being diversity-aware (Liu et al., 2023b). Lastly, Kung et al. (2023) introduce Active Instruction Tuning, a framework that identifies informative tasks to improve generalization, using *prompt uncertainty* as the utility function. These methods demonstrate a growing focus on automated measures of quality selection, often dynamically as part of the model training process.

**Data selection is increasingly guided by data governance, liability, and other concerns** Data selection is not always guided only by performance considerations. A large-scale audit of popular instruction tuning datasets shows a marked rise in datasets with non-commercial terms and licenses (Longpre et al., 2023b).

---

[16] https://huggingface.co/datasets/teknium/GPT4-LLM-Cleaned

Whereas non-commercial restrictions applied to $\leq 30\%$ of datasets in the prior decade, 61% of 2023 datasets had such restrictions. Similarly, new lawsuits (Saveri et al., 2023) and pushback from creators (Chayka, 2023) over alleged copyright infringement has become a central concern for developers navigating legal risk in their training data selection. The legal outcomes and enforceability of such restrictions remain uncertain, and may vary by jurisdiction. Additionally, data selection may be guided increasingly by factors related to multlinguality, target task type, or domain, as prior work has shown no size fits all for instruction tuning data (Wang et al., 2024). To address some of these concerns, the Data Provenance Initiative (Longpre et al., 2023b) has provided tools to search and filter for datasets according to license, terms, and characteristics criteria.

# 5 Data Selection for Preference Fine-tuning: Alignment

Various alignment methods, which include Reinforcement Learning from Human Feedback (RLHF), RL from AI Feedback (RLAIF), and Direct Preference Optimization (DPO), involve the integration of human preferences into model behavior. This training process is designed to steer model responses to be generally more helpful and less harmful, whereas in other training phases such as pretraining or instruction-tuning, these preference signals may not be clearly defined within the utility function. These methods, grouped under Preference Fine-tuning (PreFT), typically follow instruction-tuning in the training pipeline of large generative models. The format of this data is often trios of `(Prompt; Chosen response, Rejected`

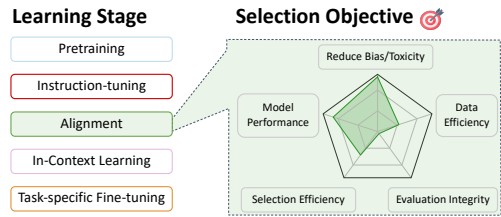

Figure 6: **Data selection: alignment.** Often, the most important objectives when selecting data for alignment is *reducing bias and toxicity* and *model performance.*

`response)`, where a `Prompt` is an instruction or other request from a user, `Chosen` is the preferred answer and `Rejected` is the less-preferred answer. Some methods work on *groups* of responses to a prompt, but they are less prevalent, and recently, Ethayarajh et al. (2024) proposed Kahneman-Tversky Optimization (KTO) which requires only the `chosen` *or* `rejected` continuation, but not both. The benefit of PreFT methods is to remove the requirement of strict next-token or accuracy-based predictions from ML models and integrate other, sometimes nuanced, qualitative signals instead.

RLHF and RLAIF methods include the important step of independent training of a reward model for active selection or weighting of samples during training. A reward model is a separate model fine-tuned to take in any input, such as text, and output a scalar reward indicating the utility (as represented by the probability that the text would be chosen). Data selection methods can be applied via training a reward model or the downstream policy, except for Direct Preference Optimization (DPO) methods that directly optimize the reward model and policy jointly from one set of data (Rafailov et al., 2023).

Data selection methods for PreFT are very nascent and are often focused on obtaining signal on specific capabilities and evaluations from the model. In this setting, the primary approaches to data selection are manual filtering, model-based evaluation, and reward model re-weighting (*e.g.*, rejection sampling). The simplicity of the data selection methods reflects the relative underspecification of the goals of preference training, where basic questions are actively being studied on how best to collect preference data, and for what purpose (Lambert et al., 2023a; Casper et al., 2023). This area is also studied in Liu et al. (2023b) primarily for aligning instruction-tuned models, where the authors focus on three axes: complexity, quality, and diversity.

**Manual and heuristic selection.** Initial methods for collecting and filtering data for PreFT involved manually selecting data and using heuristic utility functions. Multiple works have indicated the use of external organizations to procure their data (Touvron et al., 2023b; Nakano et al., 2021; Bai et al., 2022b; Rajani et al., 2023), where detailed instructions are written by researchers and sent to a data-labeling company. However, it is common practice for data-labeling companies to filter the data internally before sending it to the model training organization and these processes are not documented beyond general notions of controlling for diversity and quality. Furthermore, this process is not reproducible. Ethayarajh et al. (2022) filter Reddit data based on the number of votes received on comments and posts in the comment chain when creating the Stanford Human Preferences (SHP) dataset. Similarly, Bai et al. (2022a) and Lambert et al. (2023b) weigh data points based on a measure of minimum engagement from users when creating their respective datasets. `UltraFeedback`, a popular dataset used to train state-of-the-art chat models such as Zephyr-$\beta$ (Tunstall et al., 2023) and Tulu 2 (Ivison et al., 2023b), implements a multi-aspect filtering process to create a preference dataset focused on instruction following, helpfulness, truthfulness, and honesty (Cui et al., 2023a). Further manual filtering has been performed on this data, and others such as Orca DPO pairs[17] improve the performance by verifying accuracy (Bartolome et al., 2023). The authors perform stratified

---

[17]https://huggingface.co/datasets/argilla/distilabel-intel-orca-dpo-pairs

sampling across a variety of existing preference datasets, filter for length, and decontamintate their data. Additionally, they use a set of 17 models to obtain diversity in their dataset.

**Model-based selection.** While human oversight can be higher quality, the ability of models to accurately label undesirable (*e.g.*, incorrect) data has improved significantly, leading to an increased usage of language models to assign utility. For example, the Phi series of models (Gunasekar et al., 2023) use filtering to achieve "textbook quality" data from the web, as performed by a custom LM-based classifier, combined with synthetic data generated from OpenAI models. `UltraFeedback` (Cui et al., 2023a) and `Nectar` (Zhu et al., 2023) use GPT-4 to filter and/or rank the responses for quality. Critic models (Shepherd (Wang et al., 2023a) and Prometheus (Kim et al., 2023)) are used to critique model responses and identify diverse errors, and can be used to rank data points for quality. An example of using a model to filter data is by asking an LLM if an edited piece of text fixed a critique, and if the answer is no, the datapoint is removed, as done in Castricato et al. (2024). They are used as a part of the RLAIF cycle, where the critic is used as a reward model, to rank multiple model outputs. As this area of methods matures, it is likely that such critic models will form a central part of the data selection process by simply removing erroneous data.

**Reward model re-weighting.** A number of works have also demonstrated that using a reward model directly to assign utility to candidate data points is a viable data selection method (Bai et al., 2022b; Touvron et al., 2023b; Yuan et al., 2023). In rejection sampling (RS), or Best-of-N sampling (BoN) as used in Nakano et al. (2021); Gao et al. (2023a), $n$ candidate outputs are sampled from the generative model and the reward model selects the best candidate. Touvron et al. (2023b) use rejection sampling as part of a continued instruction-tuning by selecting the top percentage of their instruction dataset to continue training on, while Yuan et al. (2023) use rejection sampling over reasoning paths to improve mathematical abilities, where reasoning path length is a key factor. They evaluate the reasoning steps programmtically in a Python environment and use the candidate completions from multiple models while fine-tuning a single model. Liu et al. (2023a) improve on standard rejection sampling, inspired by the unsupervised RL training loop, to have a critic model consistently evaluate data in the RLHF process, gating which data is passed to the model and offering critiques on the rejected data. Beirami et al. (2024) demonstrated that a popular KL distance measuring the impact of Best-of-N sampling is actually bounded by KL distance rather than a scaling law on the number of samples scored over.

Pace et al. (2024) propose West-of-N sampling, a method of utilizing synthetic data to improve the reward model, which can in turn improve the base generative model with RS or BoN techniques. West-of-N sampling takes the synthetically generated candidate outputs (as in rejection sampling) and extracts the best and worst candidates as a pair of preference data. This data is then used to iteratively fine-tune the reward model, which can be used for many of the other methods in this section. Similar to West-of-N is the method of Self-rewarding LLMs (Yuan et al., 2024b), where instead of using the synthetic pairwise data to train a reward model, the data is iteratively used to directly update the policy with direct preference optimization (DPO) (Rafailov et al., 2023). However, self-rewarded models have been shown to be biased towards their own rewards (Xu et al., 2024a) and, similarly to other methods which utilize synthetic data, it is crucial to ensure that the data generating process yields a diverse set of data to reduce the chances of mode collapse.

Emerging methods have applied reward-weighted data selection within a feedback loop, more akin to earlier forms of RLHF, such as with DPO models (Yuan et al., 2024b), or methods imitating self-play with less clear data selection methods (Chen et al., 2024). For example, Liu et al. (2024b) performs a rejection-sampling-like optimization with statistical weightings predicting the likelihood of whether a completion came from the optimal policy or base SFT policy to improve downstream preference optimization similar to DPO or Sequence Likelihood Calibration (SLiC) (Zhao et al., 2022). Especially in preference-based alignment, the border between synthetic data generation and data selection are often blurred as best practices are determined.

# 6   Data Selection for In-Context Learning

In-context learning (ICL) is a widely used prompting paradigm for language models (LMs). Instead of being used to fine-tune the LM, a few demonstration examples are given as a prompt to guide the language model to perform a similar prediction task on the input query (Brown et al., 2020a). In-context learning is known to be sensitive to the choice (Perez et al., 2021), and even ordering (Lu et al., 2022), of the demonstrations. To improve the ICL performance without training the potentially large LM, many recent papers have worked on constructing better in-context demonstrations by: selecting an optimal ordering from a fixed set of demonstrations, selecting from a large set of labeled data, or strategically annotating a small set of unlabeled data.

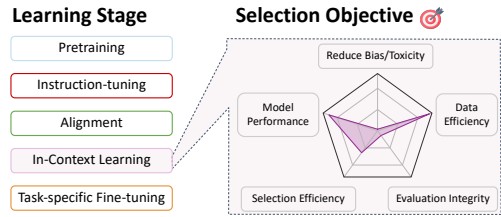

Figure 7: **Data selection: in-context learning.** Generally, the most important selection objectives for in-context learning are *data efficiency* and *model performance.*

**Demonstration reordering.**   Lu et al. (2022) point out that the ordering of demonstrations significantly affects the in-context learning performance, and propose to reorder the demonstration using the entropy of the predicted labels as a utility function.

**Demonstration selection.**   Suppose we have a set of annotated data that cannot all fit into the prompt of a pretrained language model, and we also do not want to fine-tune the potentially very large language model due to a limited budget. How do we select the set of data points to be used as in-context demonstrations that leads to optimal performance? As discussed below, some methods select a fixed set of demonstrations, while others select demonstrations for each individual test data point using either retrieval-based methods or sampling-based methods.

Some works have focused on **selecting a fixed set of demonstrations** that achieves the best average performance across all testing inputs sampled from a target task. For example, Zhang et al. (2022a) use reinforcement learning techniques to learn a policy that assigns utility to demonstrations for use on previously unseen tasks. A different approach, proposed by Wang et al. (2023b), is to use a smaller pre-trained language model to learn task-defined latent tokens and then select the examples that can best reconstruct the latent tokens.

Many other works propose **retrieval-based methods**, where they train a demonstration retriever that can select demonstrations specifically tailored to each individual test data point. A straightforward method is retrieving the most similar examples to the testing input by cosine similarity of sentence embeddings (Liu et al., 2022). Gao et al. (2023b) further enhance this approach by retrieving demonstrations whose label lies in top-2 zero-shot predictions of the testing input. Rubin et al. (2022) score the similarity-based retrieved candidate demonstrations by their one-shot in-context learning performance for sequence-to-sequence tasks, then train a retriever with the collected scores. Li et al. (2023c) advance this framework through unified training across various datasets. Hashimoto et al. (2023) analyze the effect of different utility functions in training demonstration retrievers, and propose that incremental utility, which is the 1-shot utility score minus the 0-shot utility score, is better than directly using the 1-shot utility score. Additionally, Iter et al. (2023) show that the cross entropy difference between candidate demonstrations and the test example is a promising selection utility. Ye et al. (2023) improve upon the previous works by scoring a set of demonstrations instead of only one demonstration. Finally, when prompting a model for semantic parsing, Levy et al. (2023) propose to select diverse demonstrations that collectively cover all of the structures required for the output program. For a comprehensive survey on this line of work, please see Xu et al. (2024b).

There are also (iterative) **sampling-based methods** that do not require training a retriever to select the demonstrations. The most straightforward approach would be to uniformly sample a set of demonstrations from the candidate set as used by Min et al. (2022). Chang & Jia (2022) train a datamodel (Ilyas et al., 2022) to predict the validation performance for a set of chosen demonstrations and select the demonstrations with the

highest validation performance. Similarly, Nguyen & Wong (2023) propose constructing a validation set and evaluating each training data point by contrasting the validation performance of prompts with and without that data point, and formalizing the performance difference as influence. Li & Qiu (2023) iteratively select in-context demonstrations by an informativeness score, which is the language model prediction probability difference between one-shot and zero-shot inputs, and a diversity score, which discounts semantically similar examples. Gupta et al. (2023) propose to select the most informative set of demonstrations, instead of individual informative demonstrations, with a greedy algorithm to maximize the coverage of the chosen demonstrations. Similarly, Xu & Zhang (2024) iteratively refine the selected set of demonstrations by computing the misconfidence score, which is the ratio between the highest probability that the language model assigned to any incorrect label, and the output probability of the correct label.

**Selective annotation.** SU et al. (2023) first propose the selective annotation setting: given an unlabeled dataset and a fixed budget, the goal is to select the most informative examples for annotation, which are used to further select demonstrations for each testing input to maximize ICL performance. In this way, we aim to minimize the human annotation effort in creating labeled datasets. We also maximize the advantage of in-context learning that only a small number of labeled examples are needed as demonstrations.

Various methods for selecting the annotation set have been proposed, usually based on the difficulty and diversity of the unlabeled examples. Then, a sentence-transformer (Reimers & Gurevych, 2019) is usually employed to compute the cosine similarity between each annotated example and the testing input, then the most similar examples are selected as the ICL demonstrations.

SU et al. (2023) propose VOTE-K, which constructs a graph by applying k-NN over the sentence-transformer-embedded (Reimers & Gurevych, 2019) unlabeled examples, to select examples for annotation. They also try to use an exponential score to discount examples that are close to the already selected examples. Zhang et al. (2023b) also contract a graph over the unlabeled examples by k-NN. They use a diffusion process to quantify the influence of unlabeled subsets and a greedy algorithm to conduct the final selection. Their proposed method, IDEAL, achieves better performance under lower time consumption than VOTE-K. Mavromatis et al. (2023) combines diversity and uncertainty sampling, by assuming each wrongly predicted demonstration is most useful in predicting inputs from its semantic neighborhood, therefore formalizing demonstration selection as a max coverage problem. Wu et al. (2023) propose an interactive data annotation system, that iteratively samples unlabeled examples with task-specific patterns and then let the human annotators correct the LLM-suggested annotations.

# 7  Data Selection for Task-specific Fine-tuning

Fine-tuning a model on a specific target task is a very different learning setting from pretraining, instruction-tuning, or RLHF, but the data selection methods that apply are not vastly different. In some ways, selecting data for a specific target task can be easier than the previous settings. First, because there is only one target task, the target distribution will generally be more narrow than in either pretraining, instruction-tuning, or multi-task learning. Also, task-specific fine-tuning is generally easier to evaluate because the target distribution is more narrow, the expected use cases are more clear, and success has a more straightforward definition, leading to a less ambiguous evaluation than the previously discussed settings.

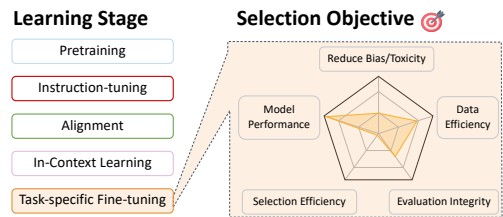

Figure 8: **Data selection: task-specific fine-tuning.** Commonly, the most important objectives when selecting data for fine-tuning are *model performance*, *data efficiency*, and *evaluation integrity*.

Data selection for task-specific fine-tuning can be roughly divided by whether the goal is to match the target distribution or to diversify the existing data distribution. The first setting, where the goal is to match a target distribution, is particularly beneficial in data-limited situations such as few-shot learning (Albalak et al., 2022b; Ivison et al., 2023a; Albalak et al., 2023b). For example, there may be very little data for the target task (the task we want the model to perform), but we do have access to a wide variety and large quantity of auxiliary data that we can utilize. The second setting, where the goal is to diversify the data distribution, can further be divided into two settings, where the goal is either to improve data efficiency (Lang et al., 2022; Maharana et al., 2024), or improve the model's robustness (Haghtalab et al., 2022; Sagawa et al., 2020; Bejan et al., 2023).

**Selecting from auxiliary data to improve generalization.** There have been a number of works that utilize auxiliary data (i.e. data from a source other than the target task) to improve a model's ability to generalize to the target distribution. In these settings, the goal of data selection is to determine the optimal auxiliary data that improves a model's performance on the target distribution, and because we have access to data from the target distribution the utility function for these methods is often a form of similarity between the auxiliary and target data. These methods tend to have some similarities with domain-specific data selection for pretraining data (§3.4) with the main difference being that in a fine-tuning setting, more computationally expensive utility functions are feasible.

The availability of pretrained models has been a huge benefit to the ability of models to generalize on fine-tuning settings, allowing information learned during pretraining to be transferred and applied to the downstream task. A number of earlier works have taken this a step further and perform an intermediate training step of training on auxiliary data between pretraining and fine-tuning, often relying on empirical evidence to determine which auxiliary data is best (Phang et al., 2018; Pruksachatkun et al., 2020; Iter & Grangier, 2021; Grangier & Iter, 2022).

Phang et al. (2018) introduce the *Supplementary Training on Intermediate Labeled-data Tasks* (STILTs) method where the goal is to fine-tune a model for tasks in the GLUE dataset (Wang et al., 2018a). They manually select textual entailment (MNLI (Williams et al., 2018), SNLI (Bowman et al., 2015)), paraphrase detection (QQP[18]), and fake-sentence detection as their auxiliary tasks specifically because they are similar to the tasks in the GLUE dataset. Similarly, Pruksachatkun et al. (2020) evaluate the transfer between 11 intermediate auxiliary datasets and 10 target tasks, but also evaluate the models on 25 probing tasks to try and understand what skills are being learned from the auxiliary data. These methods provided early efforts at understanding task transfer, but the data selection was performed entirely by human decisions on similarity to the target task.

While Phang et al. (2018) and Pruksachatkun et al. (2020) focus on task transfer from the auxiliary to target data, Iter & Grangier (2021) and Grangier & Iter (2022) focus on domain adaptation from the auxiliary to

---

[18]https://data.quora.com/First-Quora-Dataset-Release-Question-Pairs

target data. Iter & Grangier (2021) find that the best data selection method for domain adaptation uses a domain classifier as the utility function, a model that is trained to discriminate between data from the target distribution and from a general distribution, similar to methods on domain-specific selection for pretraining (§3.4). Grangier & Iter (2022) further explore the use of three separate data selection methods: importance sampling, contrastive data selection, and influence functions, finding that all three methods select for very similar data points.

More recently, methods that select from auxiliary data train on both the auxiliary and target data simultaneously. For example, FETA (Albalak et al., 2022b) performs a comprehensive task transfer study similar to Pruksachatkun et al. (2020), training a model on all auxiliary-target pairs, and finding that similarity between the auxiliary and target task's label-space (*e.g.*, binary classification, multi-class classification, etc.) can be used as a good heuristic utility function. However, using the label-space is only possible when the structure of the target task is very clear, and is only useful when the label-space is fairly unique. Additionally, the structure of a task does not fully describe the data, and this method is not feasible at large scales as it requires human judgment. DEFT-FEW (Ivison et al., 2023a) improves on the ideas from FETA by using a model-based utility function to select the 500 auxiliary data points that are nearest neighbors to the target dataset. SKILL-IT (Chen et al., 2023c) also builds on the ideas from FETA (Albalak et al., 2022b), defining tasks as different skills that a model can learn, and learning a pairwise correlation between skills by training on each pair of skills to find when skill A benefits skill B. Based on this, they create a graph of skills ordered by dependencies and define a data mixing algorithm that incorporates the loss on the target task. However, training a separate model on every pairwise combination of tasks may be very inefficient. Albalak et al. (2023b) improves upon the inefficiencies of SKILL-IT by formulating the auxiliary data selection problem as a multi-armed bandit, where the utility function is an aggregated reward function of the cosine similarity and magnitude similarity between gradients of batches from the auxiliary and target datasets. By formulating the auxiliary data selection problem as a multi-armed bandit, their algorithm runs online without requiring any knowledge of the auxiliary data prior to training. While Albalak et al. (2023a) design a gradient-based utility function for data mixing, Xia et al. (2024b) use a similar gradient-based utility to assign values to individual auxiliary data points. To improve the efficiency of assigning utilities to each data point, they only calculate gradients for the parameter-efficient LoRA (Hu et al., 2022) modules and further reduce the gradient dimension through a random projection.

**Improving task-specific data efficiency.**   Another setting where data selection can be applied beneficially for task-specific fine-tuning is when we wish to select a subset of the target dataset to improve data efficiency. Methods in this category have a similar motivation to fuzzy data deduplication from §3.5, but because the target datasets when performing fine-tuning are significantly smaller than pretraining corpora, the methods used here can apply more complex, computationally expensive utility functions.

Maharana et al. (2024) propose a method of diversifying data based on the expected difficulty. They represent the dataset as a graph, where individual nodes (data points) are connected to their k-nearest neighbors in representation space and use variability (Swayamdipta et al., 2020) as the utility function for each individual data point. Next, they perform a forward message passing process which propagates information on the utility function of a data point to its neighbors. Finally, they perform data selection through an iterative reverse message passing process, where at each step the highest scoring data point is kept, and all of its neighbors get down-weighted to encourage the selection of diverse data points. Lang et al. (2022) also use a graph representation as part of their utility function, determining which weakly-labeled data points to keep based on their nearest neighbors, and the label of their nearest neighbors. They only keep data points whose nearest neighbors have mostly the same label, thereby using the representations geometry to identify accurate subsets and removing ambiguous data points. Bejan et al. (2023) develop the task-agnostic *self-influence* score as a utility function for filtering data (and designing a curriculum). The intuition behind the self-influence score is: if not training on data point $x^{(i)}$ leads to a significantly higher loss on that data point, then there must not be many supporting data points around $x^{(i)}$ and it is likely to be an outlier, and therefore a low-quality data point. They suggest that self-influence functions can identify data with label noise, out-of-distribution data, data that is ambiguous, and difficult to learn samples, and removing such data can improve the models performance.

Many methods of improving data efficiency exist in general machine learning settings (*e.g.*, data pruning and coreset selection), but most have exclusively been developed and evaluated in the computer vision domain. Azeemi et al. (2023) is one of the limited tests of data pruning on language tasks, where they observe that selecting for difficult to learn examples leads to improved model performance, while reducing the dataset size by up to 60%. The many other works on data pruning and coreset selection (further discussed in §9) may provide effective methods of improving data efficiency for task-specific fine-tuning, but could require some alterations to work for NLP tasks.

**Selecting data for robustness.** Finally, an important problem that data selection can be applied to is improving a model's robustness by ensuring that a model is not biased towards any particular subpopulation of data and to avoid relying on spurious correlations. Because these methods concern themselves with subgroups of data, they share similarities with data mixing (§3.8). Sagawa et al. (2020) introduce group distributionally robust optimization (GROUP DRO) to train models that minimize the *worst-case* loss over groups in the training data. As their utility function, they use a notion of the generalization gap (the difference between training and validation loss) and prioritize training on data from those subgroups which have a large generalization gap. Haghtalab et al. (2022) also aim to minimize the worst-case error of a model across subgroups, while also aiming to improve the data efficiency by minimizing the number of samples required for each subgroup. They formulate the multi-distribution learning problem as a zero-sum game and aim to learn the minimax equilibrium by introducing an online learning algorithm, Resampling-based Multi-Distribution Learning (R-MDL) that uses estimates of the training error and generalization error as the utility.

**Considerations for task-specific fine-tuning.** Compared with data selection for pretraining, the goals of data selection for task-specific fine-tuning are very similar: improving data efficiency, selecting a diverse set of samples, and reducing a model's bias towards any particular subgroup. However, the task-specific datasets used in fine-tuning are often small enough that methods with high computational requirements are plausible. Though many of the methods presented in this section may be beneficial for pretraining (or other settings), they may be unreasonably costly to use on web-scale data. For example, Azeemi et al. (2023) requires training a model on the full dataset prior to determining which data points to keep/remove, and Haghtalab et al. (2022) assumes that the model will be validated on each subgroup at every step of the learning algorithm. Although these ideas may be computationally prohibitive to be used as-is, it may be possible to adjust them to work at larger scales by reducing the frequency of updates, or using smaller models as proxies.

# 8   Data Selection for Other Domains

This survey focuses on data selection for language models, but data selection is an active area of research in computer vision, vision-and-language and broadly in machine learning. Some of the data selection methods covered by this survey are domain-specific (*e.g.*, the total number of characters in $x^{(i)}$), while others are domain agnostic (*e.g.*, selecting data most similar to a target distribution). Therefore, some of the methods previously presented in this work may be more or less applicable in other domains. Similar to language models, data selection methods have been applied at multiple stages of model training in other domains (*e.g.*, vision-language model pretraining) and for similar purposes (*e.g.*, data efficiency, model performance, and reducing bias). In this section we will briefly discuss some of the prominent uses of data selection in non-language domains and compare and contrast them with the previously discussed methods.

**Data selection for pretraining.**   Of particular interest has been the rise in pretrained vision-language models, which has followed closely after the success of large language models (Radford et al., 2021; Schuhmann et al., 2022; Zhu et al., 2024). For example, Schuhmann et al. (2022) and Zhu et al. (2024) design datasets for vision-language pretraining using some of the same methods and ideas presented in §3 as well as some methods that are specific to their vision-language data. Schuhmann et al. (2022) collect web-text along with images that contain an *alt-text* in order to create data with image-text pairs. They utilize some heuristic filters on the text, similar to those from §3, but they also remove data with less than 5kb of image data, as well as image-text pairs that have cosine similarity less than 0.28 for english data and 0.26 for other languages (image and text separately encoded using OpenAI's ViT-B/32 CLIP model). The similarity step removed significant quantities of data, reducing the original corpus from around 50 billion images to 6 billion. While `LAION-5B` is a corpus entirely of image-text pairs, `Multimodal C4` (Zhu et al., 2024) introduces a dataset of interleaved sentences and images, where each data point can contain multiple sentences and images. Zhu et al. (2024) use a preprocessed English-only `C4` dataset and gather image files (`png/jpeg/jpg`) that were found on the source websites for `C4`. They perform an image-deduplication step to ensure that a single image is not overrepresented within a single document or across the entire corpus. Then, they discard images with a height or width smaller than 150 pixels, which accounts for many small icons and navigation buttons, as well as images with aspect ratios greater than 2 or less than 0.5 (removes many banners at the top/bottom or sides of websites). Finally, to interleave images into the existing text, they use cosine similarity between images and individual texts, interleaving only the images which have a similarity score of at least 0.15. Another unique filtering done for this domain is a step of removing images that contain faces.

Recently, Bruce et al. (2024) collect a dataset of publicly available internet videos and train a *foundation world model* to generate action-controllable worlds given image inputs. They filter their dataset using a mixture of heuristics (*e.g.* video titles must contain certain keywords such as "speedrun" or "playthrough") and quality filtering. Specifically, they hand label 10,000 videos and train a classifier on their labeled data.

These methods mainly filter data with heuristic approaches, and have the same drawbacks as heuristic approaches for language datasets; namely, the difficulty of validating a method, and slow iteration time as it requires training multiple models. *Data Filtering Networks* aim to overcome these weaknesses by filtering data not with heurstics, but with a trained network (Fang et al., 2024). While Fang et al. (2024) demonstrate that it is possible to train a filtering network that can select data points which lead to a better model, in a cruel twist, they find that data quality is key to training a good filtering network, leading to a circular problem.

In addition to data collection and filtering, data mixing is also a topic of concern for non-language pretraining. Piergiovanni et al. (2023) propose an online data mixing method for vision-language models that mixes a variety of tasks together (*e.g.*, image captioning or visual question answering). Their method, dynamic difficulty sampling, calculates the loss of each task $t$, $\mathcal{L}_t$, and weights each task proportionally with the total loss $L$, getting the final sampling weight as $S_t = \dfrac{\mathcal{L}_t}{L}$. This mixing method maintains some similarities to the online data mixing methods from language model pretraining (Xia et al., 2024a; Albalak et al., 2023a; Schioppa et al., 2023).

**Data selection for task-specific fine-tuning.**   The popularity of task-specific fine-tuning across modalities and domains has led to many data selection methods that target this setting. For the most part, the methods of data selection for non-language domains are quite similar to the methods for language. Generally speaking, these methods use the output of a model as the representation upon which the utility function is built. In an early work on support vector machines, Wang et al. (2005) devise a method of selecting data that reduces the training set size while maintaining generalization by filtering out data points which do not affect the final decision function. This idea resonates in the language domain, with similarities to methods that use loss (Albalak et al., 2023a) or methods that attempt to estimate data difficulty (Mindermann et al., 2022). A more recent direction has been how to best utilize synthetic data, and Yuan et al. (2024a) demonstrate that generating large quantities of synthetic images and selecting data using distribution matching (matching the synthetic data to the target data using maximum mean discrepancy) leads to models that perform well for image classification with synthetic data only, as well as when mixing real and synthetic data.

**Adaptive batch selection.**   A number of methods have been developed to improve the data selection on the level of individual batches, sometimes called data difficulty estimation. One of the motivations of these methods is that they use the current model's state to determine which data to select, but this can add significant computational overhead. Additionally, because these methods are run alongside the model training, their cost is not amortized across training runs as it would be for static datasets. Chang et al. (2017) propose two methods for ACTIVE BIAS that emphasize training on data points where the model is uncertain, based on their prediction variance or closeness to the decision threshold. Similarly, Song et al. (2020) propose RECENCY BIAS, a method that further emphasizes training on uncertain data points that were predicted inconsistently in recent iterations of training. Finally, Mindermann et al. (2022) present *Reducible Holdout Loss Selection* (RHO-LOSS), a method of adaptive batch selection that, intuitively, avoids selecting data points that are redundant, noisy, or less relevant. RHO-LOSS was validated on a very small sample of NLP tasks, task-specific fine-tuning on `SST2` (Socher et al., 2013) and `CoLA` (Warstadt et al., 2019), but has not been demonstrated to work in other learning settings, and Kaddour et al. (2023) show that it is not computationally efficient.

Adaptive batch selection methods are similar to online data selection, except that they are often performed on the level of individual data points, rather than groups, domains, or tasks. This particular direction has found minimal success with LLMs, in particular for pretraining (Kaddour et al., 2023), but the success of such methods in other domains suggests that it would be worthwhile adapting these methods to language.

**Bias, fairness, and robustness.**   Reducing a models bias towards or away from any particular class is a long-standing challenge in machine learning, related to the desire to create fair datasets and improve a model's robustness to distribution shifts and out-of-distribution data.

There have been a number of recent works that aim to prepare fair datasets. Biswas & Rajan (2021) analyze stage-specific fairness (SF) which measures fairness metrics with and without certain data preprocessing stages. González-Zelaya et al. (2023) introduced FAIRPIPES, a heuristic approach to find an optimal data preparation pipeline configuration. A number of works have also developed software tools for tracing distribution distortions linked to fairness in the data preparation workflow (Yang et al., 2020a; Grafberger et al., 2021; Schelter et al., 2019).

Additionally, there exist fairness-aware balancing techniques (Yan et al., 2020; Sonoda, 2023; Chakraborty et al., 2021; Salazar et al., 2021; Yang et al., 2020b) as well as reweighting/resampling methods aimed at ensuring fairness (Kamiran & Calders, 2012; Jiang & Nachum, 2020). In NLP, using balanced training for fairness has been explored (Wang et al., 2019; Han et al., 2022). Jeong et al. (2022) showed that different proportions of demographics can lead to nontrivial differences in fairness. To alleviate the challenge of gathering additional data, recent research has focused on designing adversarial networks. These networks are developed to acquire a balanced representation, which helps in minimizing the adverse effects of sensitive attributes (Madras et al., 2018; Adel et al., 2019).

Learning representations that are robust to distribution shifts, or out-of-distribution data, is very challenging, and a very active area of research. Deng et al. (2023) propose *Progressive Data Expansion* (PDE), a method for learning robust representations by progressively expanding the training dataset. While naively training on

all data creates models that are susceptible to learning non-generalizable spurious features, by progressively expanding the training dataset, Deng et al. (2023) demonstrate that PDE improves the model's robustness for a better worst-group performance. Nguyen et al. (2022) demonstrate, on six publicly available image-text datasets, that each individual dataset can lead to robustness to certain distribution shifts, but no individual dataset dominates the robustness. They explore the interactions between datasets by combining multiple datasets and find that this does not lead to an overall better model, rather it dilutes the robustness from each component dataset. Their results demonstrate that simply gathering large amounts of data is not optimal.

**Increasing the rate of data selection research.**    Recently, the importance of data selection and quality has gained notoriety, and thus a few efforts have been developed to try and lower the barrier to entry for research to be done. One such effort is `DataComp` (Gadre et al., 2023), a benchmark for multimodal dataset design. `DataComp` provides two tracks for participation: the filtering track, and the bring-your-own-data track. The filtering track is of particular interest as it gives participants a common dataset (12.8B image-text pairs from `Common Crawl`), and the participants' goal is to determine the best subset to train a model on. Additionally, Gadre et al. (2023) provide over 300 baseline experiments, and find that, similar to the language-only setting, a smaller, more selective dataset can lead to models that generalize better than larger datasets. Of course, this still comes with the same limitations as all previously discussed sections, where the exact evaluation setting plays a large role in what data is "best". Specifically, `DataComp` evaluates models' abilities to perform image classification and retrieval tasks, but does not evaluate generative capabilities. This suggests that distribution matching methods may fare well on `DataComp`, while an evaluation of generative capabilities may prefer data selection methods that favor distribution diversification. `DataPerf` (Mazumder et al., 2023) provides another entry for data selection research. The `DataPerf` benchmark contains 4 data selection settings: vision, speech, debugging, and language. The various settings allow researchers to test their algorithms for selection, augmentation, quality assesment, and cleaning. Mazumder et al. (2023) find that automated methods tended to perform better than manual cleaning or filtering; specifically, they found that submissions succeeded with automated methods for recognizing noisy images and labels, identifying mislabeled images, correcting class imbalance, and selecting and enhancing images from the long-tailed distribution of classes.

By providing a shared resource where researchers can test their methods, benchmarks and competitions like these can help to increase the pace of research, as well as giving a direct comparison between methods. Specifically, `DataComp` has evaluated 21 submissions[19] and `DataPerf` has evaluated 54 submissions.[20] Some of the promising methods from these benchmarks may be useful for language domains, but their transfer may be limited due to their domain-specific nature. Benchmarking data-centric methods, and data selection in particular, for the language domain has not been fully utilized and should be in the future to accelerate the pace of data selection research.

---

[19]As of 02/06/2024, found at `https://www.datacomp.ai/leaderboard.html`.
[20]Across vision, speech, debugging, and language tasks, as of 02/06/2024, found at `https://www.dataperf.org/`

## 9 Related Topics

**Data cleaning.** As discussed in sections 3.2 and 3.5, some utility functions can determine both whether an entire data point should be removed from the dataset and also whether a chunk of text should be removed from a data point. Data cleaning refers to the latter scenario. While some of the selection methods described in this work can also be used for cleaning pretraining data, there are additional data cleaning methods not covered here. For example, in supervised settings data can be labeled incorrectly and methods for detecting incorrectly labeled data often assume that such data are outliers, utilizing outlier detection methods to clean (or remove) the incorrectly labeled data (Abedjan et al., 2016; Narayan et al., 2022). Data cleaning can also refer to imputing missing values or finding inconsistencies across data points (see Li et al. (2021) for more details on data cleaning in non-language domain data).

**Dataset distillation and coreset selection.** Dataset distillation (also known as *dataset condensation*) and coreset selection (also known as *data pruning*) have been extensively studied for reducing the size of training data. Dataset distillation aims to *synthesize* a small dataset such that the model trained on that synthetic dataset achieve performance akin to one trained on the full dataset. Dataset distillation is first introduced by Wang et al. (2018b), where the model weights are treated as a function of the distilled dataset (outer optimization) and the distilled dataset is optimized (inner optimization) so that randomly initialized models trained on it with one-step gradient descent could achieve good performance. Such a bi-level meta-learning framework often incurs substantial computational costs. To improve the scalability and effectiveness of dataset distillation, subsequent research (Zhao et al., 2020; Lee et al., 2022b; Zhao & Bilen, 2021) has employed gradient matching as the supervision signal, aligning the gradients of model parameters trained on the original dataset and the distilled dataset. Moreover, researchers have proposed techniques such as feature distribution matching (Wang et al., 2022a; Zhao & Bilen, 2023; Zhao et al., 2023a), training trajectory matching (Cazenavette et al., 2022; Cui et al., 2023b; Guo et al., 2024), and kernel methods (Nguyen et al., 2020; 2021; Zhou et al., 2022).

In contrast, the objective of *coreset selection* (Mirzasoleiman et al., 2020a; Borsos et al., 2020; Killamsetty et al., 2021b; Guo et al., 2022; Killamsetty et al., 2021a) is to select a subset from the given dataset so that a model trained on it achieves optimal performance, which better aligns with the topics discussed in this paper. Many coreset selection methods design a series of scores and rank the data instances accordingly. Within this realm, geometry-based coreset selection methods selects the examples based on their distances from k-means cluster centers (Chen et al., 2012; Sener & Savarese, 2018; Sorscher et al., 2022; Xia et al., 2023). Additionally, predictive uncertainty and model loss metrics can also serve as criteria for ranking data instances (Coleman et al., 2019; Pleiss et al., 2020; Paul et al., 2021; Zheng et al., 2022).

There also exist coreset selection methods that are optimization-driven. CRAIG (Mirzasoleiman et al., 2020a) optimizes the coreset selection so that the sum of the gradients on the subsets closely match the gradients on the full dataset. This approach has been extended to handle noisy data settings in the subsequent work (Mirzasoleiman et al., 2020b). GLISTER (Killamsetty et al., 2021b) and GRAD-MATCH (Killamsetty et al., 2021a) focus on coreset selection on noisy and imbalanced datasets and propose to match the gradient of the coreset with the training of a held-out validation set. The ADACORE (Pooladzandi et al., 2022) method incorporates the second-order gradient information for better gradient matching. Notably, the techniques of coreset selection have found application in LLM fine-tuning (Liu et al., 2023b; Xia et al., 2024b). For example, the LESS (Xia et al., 2024b) method selects influential data for supervised fine-tuning based on the similarity of gradients evaluated on the coreset and the validation set.

**Data attribution and valuation.** Data valuation and attribution focus on understanding how each training data point affects a model's predictions. Methods for valuation and attribution can help to identify potentially mislabeled data, identify brittle predictions, quantify the affect of a training data point on an evaluation data point, and even quantify train-test contamination. These methods tend to be computationally expensive, although recent progress has improved their efficiency.

In recent work, Ghorbani & Zou (2019) propose DATA SHAPLEY, a principled framework for data valuation, where the goal is to quantify the value of each data point to a predictors performance. The DATA SHAPLEY

value for each data point can then be used as the utility function in a data selection method for either distribution matching or diversification purposes. Ilyas et al. (2022) present DATAMODELS, a framework used to predict a models performance on a data point $x^{(i)}$, given that the model is trained on a known subset of training data. They demonstrate that even using a linear function as the predictor, their datamodelling framework is capable of accurately predicting model outputs and counterfactuals, as well as encoding a metric for data point similarity. Similarly, Park et al. (2023) propose TRAK, a data attribution method that can make accurate counterfactual predictions, trying to answer the question "why did my model make this prediction". They show that TRAK is a plausible method for determining which training data was most impactful on a model's prediction.

**Data augmentation.** Data augmentation methods are used to generate new data by modifying the existing data points in a dataset. Similar to data selection, data augmentation also aims to shift the training distribution. Data selection is often used to improve the data coverage in domains where there is limited data (Li et al., 2022b; Chen et al., 2023a) and can be used in conjunction with existing data selection methods (*e.g.*, domain-specific selection in §3.4 or "selecting from auxiliary data to improve generalization" in §7). For example, Albalak et al. (2022a) use a policy gradient along with information from an auxiliary task to improve their main model. However, the quality of the coverage produced from augmentation may be limited, as the semantics of augmented data can diverge from the underlying distribution of language, leading to an undesirable distribution shift and possibly a decrease in performance (Li et al., 2022b; Chen et al., 2023a). Thus, care should be taken when performing data augmentation to ensure that the augmentations do not stray too far from the desired distribution. Nonetheless, data augmentation can still be a useful tool, and we refer readers to existing surveys on the many methods for augmentation in NLP (Chen et al., 2023a; Feng et al., 2021), code (Zhuo et al., 2023), and computer vision (Yang et al., 2022).

**Data curation.** Data curation is a set of processes surrounding the discovery, organization, integration, annotation, cleaning, storage and maintenance of data (McLure et al., 2014; Freitas & Curry, 2016; Thirumuruganathan et al., 2020). Some of the methods that are used for curation overlap with those from data selection (*e.g.*, cleaning and filtering), but there are also methods from data selection which are not included in data curation (*e.g.*, heuristic filters, data mixing, and filtering toxic and explicit content). Data selection is, generally speaking, a process that occurs after data curation, once a practitioner is prepared to train a model. After the data has been curated, then data selection methods can be used to create a dataset for the desired purpose.

**Curriculum learning.** While data selection aims to determine which data points a model should train or evaluate on, and how many times they may be repeated for training, curriculum learning aims to improve a model through an orthogonal goal. Curriculum learning is often motivated by the idea that the training data should be increasing in complexity according to the models current capability, and aims to determine *when* a data point should be trained on. Curriculum learning has found limited adoption in the natural language domain. For example, Graves et al. (2017) use multi-armed bandits to determine an appropriate curriculum for language modeling with LSTMs. Xu et al. (2020) demonstrate that curricula can be used to improve learning in natural language understanding tasks for a BERT model. More recently, Fan & Jaggi (2023) propose *irreducible curriculum* for language model pretraining, which prioritizes training on samples with higher learnability (motivated by RHO-Loss (Mindermann et al., 2022)). Curriculum learning is one aspect of data-centric AI that has not yet seen a large adoption in large language models, in part due to its limited success.

**Active learning.** Active learning is a long-studied problem that aims to answer the question: given unlabeled data points, how should a fixed annotation budget be used to optimally train a model for supervised tasks, and has been applied to a wide variety of settings ranging from text classification (Lewis & Gale, 1994) to in-context learning (Wu et al., 2023). Active learning is closely related to data selection as both fields consider the question of how to find the optimal data to train a model, but focus on different settings. Unlike data selection, active learning explicitly focuses on supervised training. Please see the available surveys on active learning in NLP (Zhang et al., 2022b), and general deep learning (Ren et al., 2021; Zhan et al., 2022; Tharwat & Schenck, 2023) for a deeper discussion on the topic.

# 10    Discussion

## 10.1    Test set decontamination

Understanding a model's capabilities requires evaluation on a fair benchmark, but the training of language models on web-scale data has raised concerns and speculation about the possible inclusion of evaluation benchmarks in training data. Given the scale of data it is not feasible to manually check, so automated methods have been designed to detect the existence of evaluation data in the training set (Rae et al., 2021; Li et al., 2022a; 2023b; Magnusson et al., 2023; Elazar et al., 2024), as well as detecting models that have already been trained on evaluation data (Carlini et al., 2018; Jagielski, 2023; Marone & Van Durme, 2024). Jacovi et al. (2023) on the other hand, suggest three strategies for reducing the risks of contamination. Data decontamination is an important step in ensuring the integrity of model evaluation, and is included in the discussion section because this critical step is relevant, and should be required, for all stages of model training.

**Detecting test set contamination with n-grams.**   Test set contamination methods often follow a similar form to deduplication methods. When decontaminating `MassiveText`, Rae et al. (2021) compute the 13-gram Jaccard similarity between train and test documents (as originally proposed by Lee et al. (2022a)) and remove train documents that have a Jaccard similarity over 0.8, the same exact method that they use to detect duplicated documents. Similarly, Brown et al. (2020a) and Gao et al. (2020) use 13-gram overlap filtering to detect contamination in their respective training sets. To make decontamination easier, some tools have been developed. For example, CarperAI released software for decontaminating evaluation data from a training dataset using MINHASHLSH.[21] Additionally, Marone & Van Durme (2024) propose *Data Portraits*, a method of investigating the data that a model was trained on. Data portraits are an artifact that records training data, allows downstream inspection, and enables answering questions about test set leakage and model plagiarism through a novel membership testing tool, a strided bloom filter. In addition to the new method, they release a demo of their tool[22] which uses 50-character grams to detect overlap between training data and an input sequence. Elazar et al. (2024) indexed several corpora using *elastic-search*, a reverse index, and used it to detect exact matches of multiple test-sets used for evaluation. Finally, Merrill et al. (2024) built an efficient automtaton named CDAWG (Inenaga et al., 2005), that allows a flexible n-gram match to the longest suffix that matches an input string in the (indexed) training data.

**Protecting test data with canaries.**   A canary string is a unique sequence of characters that is included within a dataset, or a document, used to empirically evaluate, or audit, whether a language model was trained on that document (Carlini et al., 2018). By including canary strings within a benchmark, they can protect the integrity of an evaluation metric, demonstrating the likelihood of a model having been trained on that evaluation benchmark. For example, `BIG-bench` (Srivastava et al., 2023) includes a canary GUID string in each task to allow for detecting whether models were trained on the evaluation data.[23] To detect canaries, Carlini et al. (2018) propose the *secret sharer* framework for efficiently extracting unique, secret sequences such as personally identifiable information, and demonstrate that such information can be extracted from a model. Jagielski (2023) expand on the secret sharer framework, and demonstrate how to interpret canary exposure by relating it to membership inference attacks and differential privacy.

**Detecting models trained on test data.**   Proving that black-box models have trained on a data point can be very challenging, but also important to ensure that their results can be trusted. Because their data is not disclosed, it is impossible to know with certainty what their training data was, but some methods have been proposed as efforts to address this issue. Shi et al. (2024) propose min-k% prob, using the intuition that unseen examples are more likely to contain outlier words with low probabilities according to the model, whereas unseen examples are less likely to exhibit such low probability words. While this is not explicitly a method of decontamination, it is a useful tool for determining how contaminated a pretrained models data was. This tool can be used for benchmark example contamination detection, privacy auditing for machine unlearning, and copyrighted text detection. Oren et al. (2024) present another method for detecting

---

[21]CarperAI decontamination GitHub repository
[22]https://dataportraits.org/
[23]`BIG-bench` canary GUID README

contaminated data based on the observation that models have a tendency to memorize the order of examples as they exist in the dataset, and provide provable guarantees of test set contamination through a framework of statistical tests.

**Decontaminating code data.**   Decontamination in code data is particularly important, as Li et al. (2022a) suggest that competitors in coding competitions tend to publish their solutions online after a competition. One solution to this problem is to use a temporal split when dividing data into training and evaluation. To ensure that test data does not exist in the training dataset, the training dataset can be constructed only from files that originate prior to the test competition. This ensures that the training data includes only information that would be available for human participants of the competition. In a more traditional method, StarCoder (Li et al., 2023b) decontaminates `The Stack` by removing files that contained either exact matching docstrings or solutions from the `HumanEval` (Chen et al., 2021) or `MBPP` (Austin et al., 2021) datasets, docstrings from `APPS` (Hendrycks et al., 2021), questions from `GSM8K` (Cobbe et al., 2021), or prompts from `DS1000` (Lai et al., 2023).

## 10.2   Tradeoffs between memorization and generalization

While most methods of data selection discussed here, data deduplication in particular, suggest that memorization is a negative model property, this is not always the case. There are truths about the world which always hold and memorization is indeed a desirable property in those cases. For example, when answering the question "what is the formula for Pythagoras' theorem?" we expect a model's answer to deviate very minimally from "$a^2 + b^2 = c^2$" and a serious deviation can have real world consequences if such a model was used in an education setting. However, generalization is still desirable in this situation as changing the variable names should still be recognized as correct (*e.g.*, "$s_1^2 + s_2^2 = s_3^2$"). Furthermore, memorization is also desirable for code data, such as specific API calls in tool usage for LLMs (Schick et al., 2023; Pan et al., 2023), and syntax for coding languages (Chen et al., 2021; Li et al., 2022a; Allal et al., 2023). Memorization is desirable in many other contexts as well, and Brown et al. (2021) have theoretically proven that in many cases memorization is a requirement of any reasonably accurate training algorithm.

On the other hand, it has been demonstrated that not only do models memorize the desirable information, but also the undesirable information such as personally identifiable information, phone numbers, credit card information and much more. Carlini et al. (2020) demonstrate this with a simple attack for extracting verbatim sequences of an LLMs training data, finding that although data points from the training set do not have noticeably lower loss than test data points on average, there are some individual examples that are indeed memorized. Biderman et al. (2023a) study whether it is possible to predict which sequences from the training data will be memorized by using smaller models, or partially trained models. Carlini et al. (2023) demonstrate that memorization grows significantly with three factors: the capacity of a model, the number of times a sample is repeated in the training data, and the number of tokens used to prompt the model to emit the memorized data. While the third factor, the number of tokens in a prompt, is not in the control of model developers, the first two are. Model developers can choose to use smaller models to satisfy the needs for a given use case, rather than relying on massive models which generalize across many use cases, but tend to memorize data. Furthermore, deduplication has been demonstrated as a strong method for minimizing memorization. However, as previously discussed, deduplication needs to be intelligently applied so that "good" memorization can occur (*e.g.*, facts), while reducing "bad" memorization (*e.g.*, PII).

## 10.3   There's no free lunch for filtering

Overly filtering the data can lead to undesirable biases in the model. In particular, Dodge et al. (2021) find that blocklist filtering used to create `C4` (Raffel et al., 2020) disproportionately removes text written by, and about, minority individuals. Additionally, Welbl et al. (2021) find that removing toxic content (as determined by the Jigsaw Perspective API[24]) can lead to worse evaluation loss and disproportionately affects texts about and by marginalized groups. This finding was also replicated by Longpre et al. (2023c). These

---

[24]https://perspectiveapi.com/

works demonstrate the need for filtering strategies with improved precision (remove less text that is desirable) and recall (remove more text that is undesirable).

## 10.4 Tools for data selection

When performing data selection, a first step should generally be to understand the raw dataset, and a number of tools have been developed to assist in data exploration and auditing. For example, researchers at AI2 developed a search engine over the `C4` dataset,[25] and Piktus et al. (2023) create a search engine over `C4`, the `Pile`, `ROOTS`, and text captions from `LAION`.[26] Elazar et al. (2024) describe their *What's in my big data* tool, which can search for and count n-grams in a number of popular pretraining corpora. Additionally, their online demo contains tools for exploring the internet domains included in datasets, a visualization of the amount of text overlap between corpora, and a visualization to understand the length of documents in each corpus.[27] Finally, Liu et al. (2024a) developed the INFINI-GRAM which uses suffix arrays as a method for language modeling, ultimately demonstrating competitive performance on the `RedPajama`, the `Pile`, and `Dolma` corpora in addition to being able to quickly search for n-grams within any of those corpora. [28]

Once the data has been explored and audited, the next step of data selection is to devise the appropriate selection mechanisms and apply them over the data. Some of the methods discussed in this work have been implemented as open-sourced tools, allowing anyone to easily make use of them. The CCNet pipeline (Wenzek et al., 2020) [29] is a commonly used tool for downloading and cleaning Common Crawl data. The CCNet pipeline performs deduplication at the line level, performs language identification with a fastText linear classifier, and can train an n-gram language model to calculate perplexity per document. Soldaini et al. (2024) open-source tools for identifying non-language text, language identification, personally identifiable information, toxic text, text quality, and duplicated data.[30] Xie et al. (2023b) provide a pip-installable tool for selecting data with a distribution similar to a target dataset, that can be used either for domain-specific selection, or for quality filtering.[31] Lee et al. (2022a) provide the code to run their ExactSubstr deduplication,[32] and Computer (2023) provide tools for computing exact duplicates with a bloom filter, and fuzzy deduplication with locality sensitive hashing.[33] Additionally, the tools provided by Computer (2023) contain code for calculating a wide variety of data signals including heuristics, quality, duplication, and toxicity. Additionally, CarperAI provide a pip-installable package for decontaminating datasets from a number of openly available benchmarks.[34] Finally, Penedo et al. (2024) provide a platform-agnostic python library that can filter and deduplicate data efficiently.

For a more comprehensive and up-to-date list of available tools, see the Foundation Model Development Cheatsheet (Longpre et al., 2024).

## 10.5 Considerations when applying data selection to your setting

The first step in determining appropriate actions for data selection is to determine whether you are in a setting where you wish to increase the distribution coverage from your data, or if you wish to select a subset from your data. For pretraining, the goal will usually be to increase coverage, unless the model has a well-defined downstream use case (*e.g.*, for medical domain), in which case the goal can be to select a subset of general data which is most similar to the downstream use case. For multitask training and instruction tuning, the goal is generally to increase coverage in order to ultimately produce a capable general purpose model. Additionally, depending on the number of parameters in the model you plan to train, you may have a range for the desired number of training tokens according to your compute budget (Kaplan et al., 2020; Hoffmann et al., 2022;

---

[25]https://c4-search.apps.allenai.org/

[26]code at https://github.com/huggingface/gaia, demonstration at https://huggingface.co/spaces/spacerini/gaia is not functioning as of 02/07/2024

[27]https://wimbd.apps.allenai.org/

[28]https://huggingface.co/spaces/liujch1998/infini-gram

[29]https://github.com/facebookresearch/cc_net

[30]https://github.com/allenai/dolma

[31]https://github.com/p-lambda/dsir

[32]https://github.com/google-research/deduplicate-text-datasets

[33]https://github.com/togethercomputer/RedPajama-Data

[34]https://github.com/CarperAI/decontamination/tree/main

Penedo et al., 2023). The various filter sensitivities will need to be adjusted depending on the number of tokens in the raw data in order to achieve the desired number of training tokens. For task-specific fine-tuning, the setting depends on how large the task-specific dataset is. Generally, if the dataset is relatively small, the goal of data selection will be to increase coverage over the expected test distribution. However, if the task-specific dataset is very large then models can plausibly model the distribution with fewer data points by removing redundant information from the dataset, and costs can be reduced.

The next step in the decision making depends on how much information you have about the test distribution, as this will determine which utility functions are available to you. Depending on whether you have data sampled from the test distribution, you may be able to find an appropriate model-based utility function, otherwise you will need to determine appropriate heuristics. Furthermore, the available utility functions may be limited by the amount of compute available.

Cost is another consideration when determining how to apply data filtering. Cost can be calculated in money, time, or effort. The ordering of filters used can have a great impact on the final cost. An appropriate choice of filtering order is from least expensive to most expensive, so that the more expensive selection methods will be executed on fewer data points. Additionally, Coleman et al. (2020) demonstrated that data selection methods requiring a model can be improved in efficiency by using proxy models much smaller than the actual model being trained, to further reduce costs.

## 11 Future Directions: Challenges and Opportunities

### 11.1 Accelerating research on data selection

Research on data can be a very slow and tedious process, potentially requiring significant investments of time, compute, and human effort. Here we describe four directions of future research that can reduce the effort required and the barriers to entry for research in data selection.

**Scaling down.** One direction that can enable more researchers to participate, as well as allowing for faster iteration, is scaling down the current dataset sizes and models. For example, Kaddour (2023) demonstrate that it is possible to train a model on 745x less data, but lose only 1.9% performance on GLUE (Wang et al., 2018a) and 2.5% performance on super-natural instructions (Wang et al., 2022b). Additionally, Polo et al. (2024) show that a wide variety of evaluation benchmarks can be distilled into only 100 data points, while maintaining an accuracy within 2% of the true accuracy. One area that is still uncertain is whether the results from small models scale up to larger models, or if there is any way to predict the results on large models, based on experiments from small models. This is an active area of research.

**Metrics that directly evaluate data.** One very high impact, but challenging, way to accelerate research on data selection is to develop metrics that directly evaluate the data chosen by a selection method, thereby removing (or reducing) the need for expensive model training. Recalling the definition of data selection in §2.2, the goal of a data selection method is inherently tied to it's ability to minimize or maximize the objective function of a trained model. The dependence of data selection on model training significantly increases the iteration time for experiments, reducing the amount of experimentation that researchers can perform.[35]

Preliminary works have been done in this direction, including some works on data attribution and valuation (Ghorbani & Zou, 2019; Ilyas et al., 2022), and data measurements (Mitchell et al., 2022). Works on data valuation and attribution (discussed in §9) attempt to better understand how each data point affects a model's predictions, but may be used as inspiration for methods that directly evaluate the data itself. Additionally, there are a number of metrics that directly measure the intrinsic characteristics of data including distance, density, diversity, tendencies (mean, median, mode), and association (correlation, mutual information), but these measures have yet to be connected with a model's performance on downstream data (Mitchell et al., 2022).

Developing metrics that directly evaluate data can significantly decrease the time spent on method development and increase the amount of exploratory work possible. Furthermore, removing the need to train models (which can be very costly) can allow for lower-resourced organizations and individuals to contribute to the field, improving diversity and inclusion in the process of large model development.

**Data-centric benchmarks and challenges.** Another high-impact direction for increasing the rate of research on data selection is the development of data-centric benchmarks and challenges. One hurdle of advancing data selection is that works are done across a wide variety of settings and cannot be directly compared. Specifically, much of data selection research using heuristic filters is performed as part of collecting a new dataset. This makes it impossible to compare selection methods with another dataset as they may contain different quantities of data, and sometimes even use varying model architectures and sizes. In other domains, such as vision and vision+language, challenges and benchmarks have been developed that provide well-founded evaluation settings that allows researchers from many institutions to participate in a combined effort to push the frontier of knowledge and capabilities in data selection (discussed in "Increasing the rate of data selection research" in §8). Creating similar challenges for language models can go a long way in encouraging progress.

One challenge for developing benchmarks is there is no current consensus for what the appropriate toy settings are for data selection research. With the exception of the *Loose* track from BabyLM (Warstadt et al., 2023), no prior benchmarks or challenges have been created for data selection (BabyLM is also not specifically for

---

[35]For reference, training a 1 billion parameter model on 50 billion tokens takes roughly 5 days to train on 8 A100 GPUs using GPT-NeoX (Andonian et al., 2023; Albalak et al., 2023a).

data selection, but rather to improve the speed of development on architectures and hyperparameters for language models). The DATACOMP (Gadre et al., 2023) and DATAPERF (Mazumder et al., 2023) challenges can provide inspiration for creating similar challenged and future benchmarks in language.

Due to the large number of training settings (pretraining, instruction-tuning, alignment, in-context, task-specific) and goals (general-purpose, knowledge-intensive, chat, etc.) for language models, there are a wide variety of benchmarks and challenges that would be beneficial for the community. Creating challenges and benchmarks allows for direct comparison between methods, lowers the bar for entry into the field, and encourages open progress on this important task.

**Open-sourcing tools and best practices.** As the field develops in the open, new rules and best practices will emerge. Open sourcing the tools that implement these best practices is crucial. The combination of new best practices being developed in the open, and the open-sourcing of such tools can significantly reduce the overhead required to get started into research on data. This also allows researchers to focus on specific components of the pipeline, further accelerating research progress.

### 11.2 Better understanding the properties of the target distribution

Having a clear goal is crucial to defining and developing successful data selection methods. Using the terminology and conceptual framework developed in this work can improve the understanding of what data selection methods may be appropriate depending on the properties of the desired target distribution. For example, the goal of data selection is very clear for task-specific fine-tuning, but less clear for preference fine-tuning (PreFT). Because the definition of success is less specific for PreFT, developers will likely benefit from data diversification methods. For example, it may be possible to utilize existing auxiliary data for preferences by drawing inspiration from methods in §7 ("Selecting from auxiliary data to improve generalization"), or improving robustness across preference groups by utilizing methods for robustness (§7 "Selecting data for robustness" or §8 "Bias, fairness, and robustness").

### 11.3 Shifting compute time from model training to data processing

It is now well known that training data plays a very significant role in the performance of a model, therefore creating an increased desirability to spend more compute on data processing and selection (and possibly away from model training). To improve beyond what is possible only through heuristics, it may be necessary to utilize more expensive methods.

One example of how this may be accomplished is through the use of methods which have previously only been used in low-data settings (*e.g.*, alignment) and applying them to settings with greater amounts of data (*e.g.*, pretraining). For example, methods inspired by model-based filtering, or reward model-reweighting techniques can be used to improve the "quality" of pretraining data. Additionally, techniques for generating synthetic data can be used to create more diverse training data (Bradley et al., 2023) and methods previously used only for red-teaming (Samvelyan et al., 2024) can be used to generate training data for more robust models. As compute gets cheaper, it becomes more feasible to spend the cost of directly applying the existing methods. However, another avenue worth exploring is using cheap approximations for utility functions, such as computing similarity using bag-of-n-grams and hash functions rather than from model representations.

Another method for improving data that would require large amounts of compute is additive data selection. Currently, most data selection methods (especially for pretraining) remove undesirable content from data points. However, it is possible that in the future it will be beneficial to have additive data selection methods. For example, it would be possible to use a language model to infill information within a data point. One example would be to include explanations for some phrases: *e.g.*, "the basketball player wasn't able to move *due to the double dribble rule* so he passed the ball to his teammate" where the *italicized* text has been added by an infilling model.

## 12 Conclusion

In this work, we have compiled an in-depth review on the current state of data selection and present a unified framework with which to consider and compare the wide variety of methods. This survey has described the current best practices and considerations when selecting data for training a language model. Additionally, the unified framework has allowed us to define and describe some potentially fruitful future directions of research which we have highlighted throughout the paper. However, there is much more room for innovation and improvement in data selection.

### Acknowledgments

This material is based on work that is partially funded by an unrestricted gift from Google. This work was supported by the National Science Foundation award #2048122. The views expressed are those of the authors and do not reflect the official policy or position of the US government. The work of BH and SC is partially supported by the UC Santa Barbara IEE IGSB SW Impact Grant. We want to thank Luca Soldaini for sharing their figure that was adapted into Figure 4. We are also thankful to Lucy Li for sharing her summary of different filtering mechanisms used for language models of recent years. Some of the figures include icons from the Noun Project: https://thenounproject.com/.

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

# A    Heuristic Filtering Details

**RefinedWeb (Penedo et al., 2023) heuristics.**   Removes lines based on the following rules:

- If the line is "mainly" composed of uppercase characters

- If the line is composed of only numerical characters

- If the line is a counter (e.g., "3 likes")

- If the line contains a single word

Note that some of these rules are underspecified, and the paper does not contain a threshold used to determine that a line is "mainly" composed of uppercase characters. Additionally, they do not include information on how they determine that a line contains a counter.

Furthermore, Penedo et al. (2023) also edit lines if they have fewer than 11 words and match one of the following patterns:

- At the beginning of the line (e.g., "sign-in")

- At the end of the line (e.g., "Read more...")

- Anywhere in the line (e.g., "items in cart")

Note again that these rules are also underspecified as the exact patterns that were matched are not included in the manuscript.

**MassiveText (Rae et al., 2021) heuristics.**   Remove documents that do not meet all of the following conditions:

- between 50 and 100,000 words

- mean word length is between 3 and 10 characters

- symbol-to-word ratio is less than 0.1 (for the symbols "#" and "...")

- fewer than 90% of lines start with a bullet point

- fewer than 30% of lines end with "..."

- greater than 80% of words contain an alphabetic character

- contain at least two of the following "stop words": *the, be, to, of, and, that, have, with*

**C4 (Raffel et al., 2020) heuristics**   Remove lines that:

- do not end in a terminal punctuation, including periods, exclamation marks, question marks, and end quotation marks.

- have fewer than 5 words

- have the word "Javascript" because many of the scraped web pages had warning statements about Javascript being enabled

Remove documents that:

- have fewer than 3 sentences

- have an word from the "List of Dirty, Naughty, Obscene, or Otherwise Bad Words"[36]

- have the placeholder phrase "lorem ipsum"

- contain a curly bracket, "{" because it appears in many programming languages and they desired for C4 to contain only natural language

- have any of the phrases "terms of use", "privacy policy", "cookie policy", "uses cookies", "use of cookies", or "use cookies" because they found many pages to have boilerplate policy notices

**mC4 Xue et al. (2021) Heuristics**   mC4 is a multilingual dataset, strongly motivated by the methods of the C4 dataset. They compare that while C4 can remove lines which do not end in an English terminal punctuation mark, that is not possible for a multilingual corpus. Instead, they apply a line length filter, which removes pages that:

- contain fewer than three lines of text with 200 or more characters

- have an word from the "List of Dirty, Naughty, Obscene, or Otherwise Bad Words"[37]

- have a primary language confidence (determined by `cld3`[38]) below 70%

**MADLAD-400 (Kudugunta et al., 2023) Heuristics**   MADLAB-400 is a multilingual corpus, so some of the heuristics are unique to a multilingual setting, but many can be shared with English-only corpora.

They remove pages that:

- Contain fewer than 4 lines with 200 or more characters (as in Xue et al. (2021))

- have fewer than 5 sentences

Remove lines that:

- contain the word "Javascript"

- contain the placeholder text "lorem ipsum"

- contain curly brackets ("{" and "}")

Additionally, Kudugunta et al. (2023) design a set of 5 characteristics that make a document "questionable". If more than 20% of the sentences in a document have one of the following questionable characteristics, then the whole document is removed. The questionable characteristics are:

- The predicted language ID of the sentence does not match the document-level language ID

- over 50% of tokens in the sentence begin with a capital letter (only if the line has at least 12 tokens)

- the sentence has less than 20 or more than 500 characters

- Over 20% of the characters in the sentence match `[0-9{}+/()>]`

- the sentence matches a "cursed regex". A cursed regex is a set of substrings and regexes that they found accounted for a significant quantity of questionable content (fully specified in Appendix A.2 of Kudugunta et al. (2023)).

**GPT-3 (Brown et al., 2020a) Heuristics**   None listed.

---

[36] https://github.com/LDNOOBW/List-of-Dirty-Naughty-Obscene-and-Otherwise-Bad-Words
[37] https://github.com/LDNOOBW/List-of-Dirty-Naughty-Obscene-and-Otherwise-Bad-Words
[38] https://github.com/google/cld3

**ROOTS Corpus Laurençon et al. (2022) Heuristics**  Laurençon et al. (2022) collect and select data from multiple sources. First they collect a code dataset from BigQuery[39] and use the following heuristics:

- remove files with fewer than 100 or greater than 200,000 characters

- remove files with less than 15% or greater than 65% alphabetic characters

- remove files with line lengths less than 20 or greater than 1000, or a token length standard deviation of 3 or less

Laurençon et al. (2022) mention their use of 8 different filters, and describe the details of each filter in a separate document[40] but do not specify what thresholds they use for their selection mechanisms[41]. They use filters for:

- Number of words

- Character repetition ratio

- Word repetition ratio

- Special characters ratio

- Closed class word ratio

- Flagged word ratio

- Language identification prediction score

- Perplexity score

**GLaM (Du et al., 2022) Heuristics**  None listed.

**Llama (Touvron et al., 2023a) Heuristics**  To train Llama, Touvron et al. (2023a) create a corpus composed of 7 domains: CommonCrawl, C4, Github, Wikipedia, Books, ArXiv, and StackExchange. The processing of each dataset is done separately as follows (where details are provided):

- English CommonCrawl - They use the CCNet pipeline (Wenzek et al., 2020) which does not perform any heuristic filtering.

- GitHub - "low quality" files were removed based on heuristics including the line length and proportion of alphanumeric characters. Additionally, boilerplate, such as headers, were removed using regular expressions.

- Wikipedia - hyperlinks, comments, and other formatting boilerplate was removed

- ArXiv - They follow the method of Lewkowycz et al. (2022) and remove everything prior to the first section as well as the bibliography. They also remove all comments, inline-expanded definitions, and user-written macros.

- StackExchange - They remove all HTML tags, and sort the answers by score so that the highest scoring answer is closest to the original question.

---

[39]https://cloud.google.com/blog/topics/public-datasets/github-on-bigquery-analyze-all-the-open-source-code
[40]https://drive.google.com/file/d/1cCJ8sWE88TRLDAa3eHLmXO4JlkR2QzLY/view
[41]It is possible that the selection mechanisms can be determined by mapping values in `parameters_filtering_default` from this file: https://github.com/bigscience-workshop/data-preparation/blob/main/preprocessing/training/01b_oscar_cleaning_and_filtering/parameters_filtering.py, but this has not been confirmed.

**Pile (Gao et al., 2020) Heuristics**   The Pile dataset consists of 22 separate sub-domains, each domain requiring it's own preprocessing. While the main goal of the Pile project was to gather data from a wide variety of domains, they include some heuristic filters when collecting data. Code to replicate their dataset processing can be found at https://github.com/EleutherAI/the-pile.

They use the following filters for each domain (where details are provided):

- Pile-CC - jusText (Endrédy & Novák, 2013) to extract the text from Common Crawl. They decide to use jusText based on visual inspection of the outputs compared with using Trafilatura, Newspaper, Goose3, and DragNet. No details on the heuristics are specified.

- OpenWebText2 - Newspaper was used to extract text instead of jusText to maintain consistency with OpenWebTextCorpus.

- ArXiv - They use the TeX source file and pandoc to convert the files to Markdown, discarding papers that had errors during the conversion process. They remove any line that begins with ": : :", which indicates an html class in Markdown.

- FreeLaw - They use `BeautifulSoup` to extract raw text from CourtListener[42].

- NIH - Award applications were removed if their abstract was too short, or missing. They also removed some administrative boilerplate (not specified).

**HTLM (Aghajanyan et al., 2022) Heuristics**   HTLM utilizes a unique heuristic because their data maintains some of the original HTML code from web documents in a simplified form called Minimal-HTML (MTHML). MHTML removes all sub-trees of the HTML DOM that does not contain text of a specific character size (128 for text elements, 64 for lists/tables/spans), and removes all *headers, footers, copyrights, forms* and *iFrames.* Then, they combine consecutive `<div>` elements, and remove any attributes which are not `class` or `id` attributes.

This preprocessing leads to data which has a mix of HTML and text. Aghajanyan et al. (2022) only use a single heuristic filter, they remove any MHTML document that has a text-to-HTML ratio of 0.46 or less.

**HumanEval (Chen et al., 2021) Heuristics**   HumanEval is a corpus of code repositories and has different considerations when filtering data. They remove files that:

- have average line length greater than 100 characters

- have a maximum line length greater than 1000 characters

- were likely auto-generated (unspecified)

- contain a small percentage of alphanumeric characters (unspecified)

**AlphaCode (Li et al., 2022a) Heuristics**   AlphaCode also creates a dataset of code repositories for training. They remove files that:

- are larger than 1MB

- have a maximum line length greater than 1000 characters

- were likely auto-generated (unspecified)

In addition to the training dataset, Li et al. (2022a) also create the CodeContests dataset to be used as an evaluation for competitive programming problems. These files contain problems, user solutions, and test cases. For this set, they remove:

---

[42]https://www.courtlistener.com/api/bulk-info/

- submissions that do not pass any tests

- tests where less than 10% of submissions produce non-empty outputs

- submissions that pass less than 10% of remaining tests

**Minerva ([Lewkowycz et al., 2022](#)) Heuristics**    [Lewkowycz et al. (2022)](#) create a dataset of 38.5B tokens combining webpages filtered for mathematics, arXiv papers, and general purpose natural language data. To create a dataset of webpages with mathematics, they filter for pages that include the string "`<math`" or "`MathJax-Element-`". Next, they extract the mathematical content from within the following html tags:

- `<script type="math/latex">`

- `<script type="math/asciimath">`

- `<annotation encoding="application/x-tex">`

