# OpenReview forum: "A Survey on Data Selection for Language Models"
_TMLR — Accepted by TMLR_

### Review · Reviewer_zFED · 2024-04-23

**Summary Of Contributions:**

This survey paper concludes a hot area in machine learning community called data selection for language models (LMs). The survey covers many novel approaches. In particular, the paper categorizes existing data selection approaches according to the data processing pipeline, and provides comprehensive reviews for each data processing stage. The survey presents further discussions and future directions with challenges and opportunities in data selection for LMs.

**Audience:**

Yes

**Broader Impact Concerns:**

No ethical concerns found

**Claims And Evidence:**

No

**Requested Changes:**

*Major*
- Provide mathematical foundations
- Provide a discussion with the related surveys in data selection for LMs or other deep models
- Provide a summary of popular datasets used in data selection for LMs
(so far, refer to Weaknesses for detailed requests)

- Provide a big table to summarize the important works for each data selection pipeline. Although the authors cover a wide range of existing works, the current version is too wordy and hard-to-follow. A big table showing the data pipeline, technical category, and properties of popular works would be very helpful to readers in organizing the existing works.
- It would be nice to have a discussion that tells what kind of recent work shows state-of-the-art performance in each pipeline. Providing empirical tips regarding the performance would be very useful for the readers to facilitate their research.

*Minor*
- In Figure 4 (and other sections), I think “Heuristics” can be included in other categories such as language filtering or deduplication. Making the “Heuristics” as a separate category seems less convincing to me.
- (typo) In Definition 2.4, "all data possible data points" should be "all possible data points"

**Strengths And Weaknesses:**

*Strengths*
- The survey covers a variety of recent approaches in data selection for LMs, from pretraining to task-specific fine-tuning.
- In terms of the data pipeline, the survey presents a clear overview of existing work.
- The survey provides further discussion and future research direction with challenges and opportunities, facilitating the application of data selection for LMs in practice.

*Weaknesses*
- **The motivation & problem scope** is a bit confusing. The survey starts with aiming to narrow the knowledge gap from organizations with resources to process large-scale data, and says “In this work, we focus on data selection for pretraining.” (in Section 2.3.4). However, the survey covers existing works across the data pipeline for LMs beyond pretraining.
- **The link with the foundations of machine learning** is missing. The mathematical foundation for explaining why data selection theoretically works better for LMs would be helpful. The current presentation is too add-hoc in some degree. Although in Section 2, the authors formulate some conceptual framework for data selection, it is not that helpful to understand the foundation of the existing work.
- The authors lack **the discussion with related surveys** in data selection for LMs or other deep models. Although Section 9 explains some data-focused related topics, it has too many technical details. I think the authors should provide some high-level explanation of how this survey differs from other related surveys in the data selection area, which can emphasize the contribution of this survey.
- There is no mention of **popular datasets** used in practice. Providing the summary of publicly available datasets used in practice for validating data selection of LMs would be very useful for the community.

---

> ### Author Response · Authors · 2024-05-11
> **Response to zFED**
>
> Thank you for the detailed review and valuable feedback. We have revised the paper (please see the updated manuscript for updates), with new content highlighted in red, and address individual comments below.
>
> **- Regarding the 1st weakness (The motivation and problem scope)**
>
> As we mention in the final paragraph of the introduction, the main focus of this work is data selection for language model pretraining, with more concise discussions provided on data selection for other training stages.
>
> Thank you for pointing this out to us, we have revised the original manuscript (highlighted in red) to clarify this point. The sentence you are referring to now reads as: “Data selection has been studied, to some extent, in all stages of training. However, information on data selection for pretraining is most limited, and therefore we dedicate the main portion of this work to comparing, contrasting, and better understanding the methods of data selection in pretraining, with more concise sections discussing data selection for other training stages.”
>
> **- Regarding the 2nd weakness and 1st requested change (Mathematical foundations of data selection)**
>
> Thank you for the great suggestion, we are in complete agreement with you, the mathematical foundations of data selection are surely lacking in the current paradigm of large language models. The combination of (1) massive training sets, (2) billion-parameter models behaving differently from million-parameter models, and (3) the compute costs required to train models, has severely hamstrung the ability of researchers to make theoretical guarantees about much of data selection.
>
> As we mention in the abstract, recent progress has mostly been driven by empirical experimentation on large-scale datasets, which is quite expensive and limiting. The very limited studies that we are aware of which could be considered as mathematical foundations of language models do not explicitly discuss data selection. For example, [1] and [2] discuss how the reasoning ability of language models is a consequence of their pre-training data. [1] discusses how chain-of-thought reasoning is effective in autoregressive language models because of local structure within pretraining data, and [2] derives novel conclusions from known facts by aggregating reasoning paths seen in pretraining data. Additionally, [3] and [4] discuss how in-context learning is a by-product of the pretraining data. They both suggest that language models learn to implicitly infer a latent variable from the given prompt at pretraining time, as the pretraining data is generated from some unknown latent variable. Thus, at inference time, models are able to infer the task semantics from demonstrations.
>
> While these are all very interesting works, we don’t believe they can form a central foundation around which we can build the mathematical foundations for all of data selection.
>
> **- Regarding the 3rd weakness and 2nd requested change (Discussion with related surveys)**
>
> The first part of this weakness is unclear to us, we are unaware of any prior surveys of data selection for LMs, can you please elaborate on this request with examples? Regarding the discussion of data selection for other deep models, in section 8 we provide comparisons with data selection for pretraining of vision, vision+language, and reinforcement learning models. If there is something in particular that you believe is missing, we would be happy to include it, please just let us know.
>
> **- Regarding the 4th weakness and 3rd requested change (No mention of popular datasets)**
>
> Thank you for pointing this out. While it would be prohibitively long to describe the datasets used for all domains and all training stages, we have included a discussion of the most popular contemporary datasets used for research on pretraining data selection in the general domain.
> Please see section 3.9.2 (Current Landscape of Datasets For Data Selection Research) of the revised manuscript.
>
> **- Regarding the 4th requested change (Table summarizing important works)**
>
> We would be happy to create a table summarizing the important works, but with the new best practices section (3.9), a table will contain significant amounts of redundant information. If you feel strongly about this, we would be happy to add it. Please let us know if, after reading the new section, you still feel that a table of important works would be beneficial.
>
> **- Regarding the 5th requested change (Current state-of-the-art and best practices)**
>
> Thank you for the helpful feedback on this point, please see section 3.9 in the revised manuscript for our updates.

---

> > ### Author Response · Authors · 2024-05-11
> > **Response to zFED (Continued)**
> >
> > [1] Prystawki et al. Why this step by step? Reasoning emerges from the locality of experience.
> >
> > [2] Wang et al. Understanding the reasoning ability of language models from the perspective of reasoning paths aggregation.
> >
> > [3] Xie et al. An explanation of in-context learning as implicit bayesian inference.
> >
> > [4] Wang et al. Large language models are latent variable models: Explaining and finding good demonstrations for in-context learning

---

### Review · Reviewer_TCqY · 2024-04-28

**Summary Of Contributions:**

The authors present an in-depth review on the current state of data selection in natural language processing (NLP). I very much enjoyed reading this outstandingly up-to-date and comprehensive overview of the state of the art in data centric language modeling. Given the importance of this topic I'm convinced that the value of contributions that aim at comprehensive surveys in this field are hard to underestimate. This holds true for all subcommunities of the ML field, but in NLP the topic deserves special attention: due to the vast amount of data and the fact that we currently do not really understand the complexity of commonly used training data sets (in contrast to, say, computer vision or tabular data sets), the research summarized in this manuscript is more heterogeneous and difficult to structure than data cleaning/selection methods in other fields.

As demonstrated by the number of references cited, structuing the complexity of this field is a very valuable contribution.

When reading the manuscript I just had minor remarks that I'll list below.

**Audience:**

Yes

**Broader Impact Concerns:**

The authors do discuss societial impact of data set quality on LLMs, including concerns such as fairness and how it could be enforced via data set curation, as well as concerns on data protection.

**Claims And Evidence:**

Yes

**Requested Changes:**

In order to guide readers a bit better through the amount of information provided in the survey it would be great if the authors could add a bit of mid-level structure (taxonomy or illustrations/rankings of methods) as mentioned above.

**Strengths And Weaknesses:**

## Strengths

* impressive breadth of literature review
* great summaries of cited content; I felt the level of detail was just right for this purpose
* the structure of the manuscript was clear and easy to follow
* the illustrations were great and helped to guide the reader


## Weaknesses
* the focus of the survey was very fine grained; except for the high level structure given by the model development cycle of LLMs, it felt like there is not much structure between the leaves of the taxonomy of methods studied and the high level structure of the model development. Maybe it would help readers to try and have some more mid-level structure? I'm not sure whether that's possible and how the best solution would look like - but maybe there are commonalities between methods that could help to reduce the complexity? Or there could be, for each section/model development step some sort of ranking, of which methods are most popular or successful in terms of quantitative metrics?
* related to the above point, I think the authors did a great job at objectively summarizing the field with a neutral tone and view. But I guess for many readers it would be helpful to have bit of guidance or insight into what works and what did not show concistent empirical improvements? In some paragraphs the authors mention that methods did not really get traction or are not really used anymore, but maybe an overview of those methods that did survive would be helpful?
* I believe active learning is probably relevant as well and has been applied to NLP problems, too [Zhang et al., "A Survey of Active Learning for Natural Language Processing", EMNLP, 2022](https://aclanthology.org/2022.emnlp-main.414.pdf) - some references probably mention literature in that field, so sorry if I just missed that,  but it seems that could maybe fit in section 9.

---

> ### Author Response · Authors · 2024-05-11
> **Response to TCqY**
>
> Thank you for the high praise! We appreciate the feedback and are glad that you enjoyed reading our work and found it to be comprehensive and outstanding. We agree that data selection, and data-centric research in general, are crucial right now in NLP and we hope this survey can be a starting point for many new works in the field.
>
> We address the requested change and weaknesses below.
>
> **- Requested change and Weaknesses 1 + 2 (Mid-level structure and current best methods):**
>
> Thank you for the feedback, we did our best to stay very neutral when discussing the vast landscape of methods, but understand that it will likely be quite useful for the community to have a set of best practices.
>
> We had previously included some mid-level information that compares methods, but this information is spread out across the individual sections for each method.
> For example, in section 3.4 (paragraph "Comparison to data quality filters.") we discuss how methods for quality filtering have essentially spun-off from methods of domain-specific selection.
> Additionally, in Section 3 we discuss reasons why the filters are generally used in this particular order, including efficiency and the amount of data they remove.
>
> To create more mid-level structure we emphasize these connections and reiterate them in a new section (3.9) to bring these connections to attention.
> Furthermore, to clarify the methods which have been demonstrated to work best, the new section of the manuscript contains the current best practices and aims to bring additional mid-level structure by discussing the connections between methods. Please see section 3.9 in the revised manuscript for the updated section.
>
>
> **- Weakness 3 (Active Learning)**
>
> Thank you for bringing this to our attention, we agree that active learning is a related topic and important to cover within the related topics section. Please see the new paragraph on active learning in section 9 of the revised manuscript (highlighted in red).

---

### Review · Reviewer_ccCq · 2024-04-30

**Summary Of Contributions:**

This paper systematically studies the data selection mechanism in language models across different stages, from pre-training to fine-tuning. It gives a thorough review to understand the techniques of data selection, including data quality, selection mechanism and so on. Generally, it summarizes a lot of findings and methodologies about data selection and facilitates the community to better choose data selection method.

**Audience:**

Yes

**Broader Impact Concerns:**

This paper does not have broader impact concerns.

**Claims And Evidence:**

Yes

**Requested Changes:**

I think this paper does not have any major request changes. It will be better to add some suggestions or examples for the community to create datasets for training large language models, and also give some details about current datasets for selection.

**Strengths And Weaknesses:**

**Strengths**

This paper systematically and thoroughly reviews and summarizes the current technologies and challenges about data selections and presents a taxonomy to understand data selections. This paper can systematically accelerate the development of data processing and facilitate the community to better use data.

**Weaknesses**

As a survey paper, I think this paper does not have any weaknesses. It will be better if authors can also present some suggestions about the integral process (which datasets, selection methods and so on) of data selection for training large language models, and give some statistical about the current dataset situations.

---

> ### Author Response · Authors · 2024-05-11
> **Response to ccCq**
>
> Thank you for the overwhelmingly positive review! We are glad that you found our survey to be so thorough, and we hope that our work will indeed accelerate progress in the field of data selection.
>
> While we appreciate that you don’t think there are currently any weaknesses, we aim to improve this work as best we can.
> To address the suggestions you mentioned, we have added section 3.9, a new section containing **(1)** suggestions on the integral processes for data selection and **(2)** details about the current datasets used for research on data selection. Please see the revised manuscript for the new section highlighted in red.

---

### Author Response · Authors · 2024-05-11
**General Response**

We would like to express sincere gratitude to the reviewers for their extensive efforts in providing us with very valuable feedback. We appreciate the time and effort that it took to review such a comprehensive manuscript and to provide us with insightful comments. We carefully considered all of the suggestions and have incorporated them into the updated version, which we feel confident has improved in quality and usefulness.

In the updated manuscript we have highlighted all the revisions in red. We briefly mention the major revisions here, and respond to individual comments as a response to each reviewer separately.

- Expanded last sentence of section 2.3.4 for clarity
- New section: Best practices (section 3.9.1)
- New section: Current landscape of datasets for data selection research (section 3.9.2)
- Added a paragraph on active learning in section 9

---

### Comment · Editors_In_Chief · 2025-11-22

Hi authors, I just wanted to inform you that, after revisiting some previously accepted papers, we've decided to grant this paper a Featured Certification. Congratulations!

Gautam, on behalf of the Editors in Chief

---

### Decision · Action_Editor_yQhi · 2024-06-18

**Recommendation:** Accept as is

**Comment:**

The paper offers a valuable contribution by providing a comprehensive review of LLM pretraining data selection methods. Reviewers have praised the systematic and thorough nature of the survey. The authors have successfully addressed most concerns raised during the revision process. The reviewers also highlight the importance of this work in structuring the complexity of LLM pretraining data research.

**Audience:**

Yes. The topic of LLM pretraining data selection is crucial for the development of LLMs, and the paper provides valuable insights that can benefit a large number of researchers in the field. Given the concentration of knowledge within a few organizations leading LLM pretraining, this survey serves as an essential resource for a broader audience.

**Claims And Evidence:**

Yes. The authors provide a systematic and thorough survey of data selection mechanisms for LLM pretraining, including detailed descriptions of algorithms and metrics. The reviewers have noted that the paper successfully addresses key concerns during the revision, particularly regarding best practices and datasets.